# Tertiary structure and conformational dynamics of the anti-amyloidogenic chaperone DNAJB6b at atomistic resolution

Vasista Adupa [1], Elizaveta Ustyantseva[2], Harm H. Kampinga [2] & Patrick R. Onck [1]✉

DNAJB6b is a molecular chaperone of the heat shock protein network, shown to play a crucial role in preventing aggregation of several disease-related intrinsically disordered proteins. Using homology modeling and microsecond-long all-atom molecular dynamics (MD) simulations, we show that monomeric DNAJB6b is a transiently interconverting protein cycling between three states: a closed state, an open state (both abundant), and a less abundant extended state. Interestingly, the reported regulatory autoinhibitory anchor between helix V in the $G/F_1$ region and helices II/III of the J-domain, which obstructs the access of Hsp70 to the J-domain remains present in all three states. This possibly suggests a mechanistically intriguing regulation in which DNAJB6b only becomes exposed when loaded with substrates that require Hsp70 processing. Our MD results of DNAJB6b carrying mutations in the $G/F_1$ region that are linked to limb-girdle muscular dystrophy type D1 (LGMDD1) show that this $G/F_1$ region becomes highly dynamic, pointing towards a spontaneous release of the autoinhibitory helix V from helices II/III. This would increase the probability of non-functional Hsp70 interactions to DNAJB6b without substrates. Our cellular data indeed confirm that non-substrate loaded LGMDD1 mutants have aberrant interactions with Hsp70.

Protein quality control (PQC) systems are vital for maintaining protein homeostasis in cells. Molecular chaperones form a crucial part of the PQC network by interacting with and stabilizing proteins. They assist proteins in many steps of their lifecycle, i.e., from being synthesized, matured, sorted to the right place, assembled or disassembled to being degraded[1]. This ensures proper protein function and, at the same time, prevents putative gain-of-toxic events, such as the formation of protein aggregates due to aberrant intermolecular protein-protein interactions. Heat shock proteins (Hsps) form a large family of molecular chaperones that were initially discovered as proteins being rapidly upregulated to protect the cell against the adverse effect of proteotoxic forms of stress, i.e., the acute formation of protein aggregates. Nowadays, we know that most Hsps are actually expressed under non-stress conditions and that most of them are not even stress-

regulated[2]. A central component of the chaperone network is the Hsp70 system that is engaged in the lifecycle of a majority of proteins. There are Hsp70s in all cellular compartments, and they all recognize substrates in a rather promiscuous manner[3,4]. Their specificity and functionality is regulated by a broad set of so-called co-chaperones; a minimal Hsp70 machine is thought to consist of at least two: a J-domain protein (JDP) and a nucleotide exchange factors (NEF)[5]. In this paper, we focus on one member of this JDP family, isoform B of DNAJB6, i.e., DNAJB6b.

Globally spoken, JDPs (of which there are ~50 in humans) have two main functions: (1) connecting client molecules to Hsp70, by tethering Hsp70 to positions where client proteins require its activity or binding to clients and delivering them to Hsp70 for further processing and (2) stimulating the ATPase activity of Hsp70 to increase its substrate

[1]Zernike Institute for Advanced Materials, University of Groningen, Groningen, The Netherlands. [2]Department of Biomedical Sciences, University of Groningen, University Medical Center Groningen, Groningen, The Netherlands. ✉e-mail: p.r.onck@rug.nl

loading[5]. By this first function, JDPs not only steer the functional specificity of Hsp70 but, when bound to the client, also maintain this substrate in a condition that is amendable for further functional maturation or degradation, ultimately preventing the (chronic) built-up of off-pathway toxic protein aggregates. As such, DNAJB6b has been shown to be an extremely good suppressor of aggregation and toxicity of disease-associated polyglutamine proteins linked to diseases such as Huntington's disease[6–10]. Furthermore, studies have shown that DNAJB6b can inhibit the formation of Aβ aggregates (associated with Alzheimer's disease)[11,12], can maintain α-synuclein (related to Parkinson's disease) in its soluble state in HEK293T-α-syn cells[13] and block the spontaneous formation of amyloid fibers of the prion Ure2 when expressed in yeast[14,15]. As part of its normal physiological function, DNAJB6b was recently suggested to play a role in the quality control of nuclear pore biogenesis by preventing phenylalanine-glycine-rich nucleoporins (FG-Nups) from undergoing non-functional phase transitions[16]. Finally, mutations in DNAJB6b have been found to cause limb-girdle muscular dystrophy type D1 (LGMDD1)[17,18]. One feature of LGMDD1 muscle is the accumulation of protein inclusions that, amongst others, contain the TAR DNA-binding protein 43 (TDP43)[17,19] of which aggregation is also associated with frontotemporal dementias (FTD) and atrophic lateral sclerosis (ALS)[20,21].

Among the three classes of JDPs (type A, B, and C), DNAJB6b belongs to the type-B subclass[5,22]. Type-B JDPs are composed of canonical DNAJB1-like proteins, to which also the yeast Sis1 proteins belong and non-canonical members like DNAJB8 and DNAJB6b[5]. DNAJB6b is widely expressed across different tissues, with particularly high expression in muscle[18,23]. It has at least two different isoforms (DNAJB6a and DNAJB6b) and a structurally and functionally highly identical homolog, DNAJB8. Like all type-B JDPs, DNAJB6b consists of an N-terminal J-domain (the signature domain of JDPs with a His-Pro-Asp (HPD) motif required for interaction with Hsp70), a G/F domain, and a C-terminal part. For DNAJB6b, the C-terminal part consists of an S/T rich domain that is important for substrate interactions[9], and a C-terminal domain (CTD) with a yet unclear function[7,24]. However, its structure is distinct from the dimerization domain in the CTD of class A and DNAJB1-like JDPs[25,26]. Despite all of this, we yet understand little about DNAJB6b's structure-function relation. Besides binding to many amyloidogenic substrates, DNAJB6b can also form (large) oligomers that depend on its S/T rich, substrate binding domain[7,8,12,24]. In vitro, it fulfills its anti-amyloidogenic functions without evident dependence on Hsp70[8,9], yet for full function in cells, interaction with Hsp70 is required[7,9]. Interestingly, Hsp70 interactions with class B JDPs (including DNAJB6b and DNAJB8) seem to depend on autoinhibitory features that are not found in class A JDPs[24,27,28]. For DNAJB6b, the autoinhibited state was proposed based on NMR data of DNAJB6b with the S/T domain deleted (ΔS/T DNAJB6b) and suggested this to be mediated by a conserved sequence motif, DIF/DVF[24]. This motif is found in helix V of the G/F domain and is present in all members of the class B JDPs[29,30] and interacts with helices II/III of the J-domain, thus blocking the HPD for functional interaction with Hsp70. What unlocks this stage, allowing Hsp70 access to DNAJB6b and its bound substrates, remains unknown. In fact, whether the autoinhibited state truly exists for full-length DNAJB6b might be disputed as Soderberg et al.[31], by combining molecular models and structural data on full-length DNAJB6b, did not capture this.

In the current paper, we study the tertiary structure and conformational dynamics of DNAJB6b in the presence and absence of the S/T domain by using all-atom explicit-water molecular dynamics simulations. We show that DNAJB6b is a highly dynamic protein, that switches between three different states: a closed state, an open state, and an extended state. Our data reveal that helices II/III of the J-domain are indeed never exposed in both the full length as well as the S/T deleted DNAJB6b. Even more so, in all three native conformations, DNAJB6b is in the autoinhibitory state. However, LGMDD1-mutated DNAJB6b shows alterations in the interactions between helix V and helices II/III that ultimately can destabilize this autoinhibition. This is in line with the NMR data in the corresponding paper (Abayev-Avraham et al.[32]), showing that this autoinhibitory regulation is disrupted in LGMDD1 mutants of DNAJB6b such that they recruit Hsp70 in an unregulated, non-functional manner.

## Results

### Initial configuration for the all-atom MD simulations

The initial configuration of DNAJB6b for the all-atom molecular dynamics simulations is obtained using homology modeling based on four different structure predictors: Robetta, TR-Rosetta, RaptorX, and Alphafold[33–38]. Initially, the models were compared to the existing NMR structure of ΔS/T DNAJB6b[24] in terms of secondary structure and the radius of gyration ($R_g$ of the J+G/F domain), and it was found that all the generated structures were closely aligned with the NMR structure. Next, the predicted structures were cross-checked with the existing lysine-specific cross-linking mass spectrometry (XLMS) data from Soderberg et al.[31] to find the best fit. With the exception of RaptorX, all predictors identified the autoinhibitory state of DNAJB6b, while the models from TR-Rosetta predicted a β-strand in the S/T domain, contradicting the disordered nature of the S/T domain[31]. From the remaining structure predictiors (Alphafold and Robetta) the Robetta model #4 demonstrated the best fit to the XLMS data (Supplementary Table 1), which we therefore selected as the initial configuration for the simulations. We carried out all-atom explicit-water molecular dynamics simulations with a total simulation time of 12 μs (for details, see the Methods section).

### Inter− and intradomain molecular interactions within DNAJB6b

DNAJB6b is known to consist of four different domains: the J-domain (1–74), the G/F domain (75–131), the S/T domain (132–188), and the CTD (189–241), see Supplementary Fig. 1. Our simulations showed that DNAJB6b samples three different states with radii of gyration ($R_g$) of 2.1 ± 0.13, 2.6 ± 0.24, and 3.1 ± 0.52 nm (Fig. 1), corresponding to a closed, open, and extended state, respectively (Figs. 2, 3).

It was previously predicted that the G/F domain (except helix V) and the S/T domain are largely unstructured[24,39], which is also what Alphafold predicts (O75190)[37,38]. Our analysis shows that they are structured (mostly helices) for around 30% (Supplementary Fig. 9a), with the structured regions involved in various interdomain interactions that play an important role in the conformational dynamics of the three states.

Detailed contact analysis on the simulation trajectories (Fig. 2) first of all revealed that the CTD does not interact with any of the other domains in the open and extended states (Fig. 2b, c, h, i). However, in the closed state, the CTD interacts with residues 77 to 102 of the G/$F_1$ domain and the N-terminal residues of the S/T domain, making the DNAJB6b more compact (Fig. 2a, g and Supplementary Fig. 3). Furthermore, our simulations showed that the G/F domain behaves as two spatially separated domains. Calculating 1D interdomain contacts (Supplementary Fig. 3) revealed a clear boundary at residue 109, dividing the G/F domain into two separate regions, which we named G/$F_1$ (residues 75–108) and G/$F_2$ (residues 109–131).

The G/$F_1$ domain interacts with the J-domain primarily through anchor region 1 (AR1) which coincides with the autoinhibited state, and that remains present in all three states (Fig. 2). In addition, G/$F_1$ interacts with the CTD, but only, in the closed state primarily through aliphatic-aromatic hydrophobic interactions (see Fig. 2a and Supplementary Fig. 5b).

The entire G/$F_2$ domain is involved in interactions with the S/T domain in the closed and open state (Fig. 2d, e and Supplementary Fig. 3). From the 1D contact distribution in Fig. 2d, e, it can be deduced that the peaks in the interdomain contacts coincide with the location

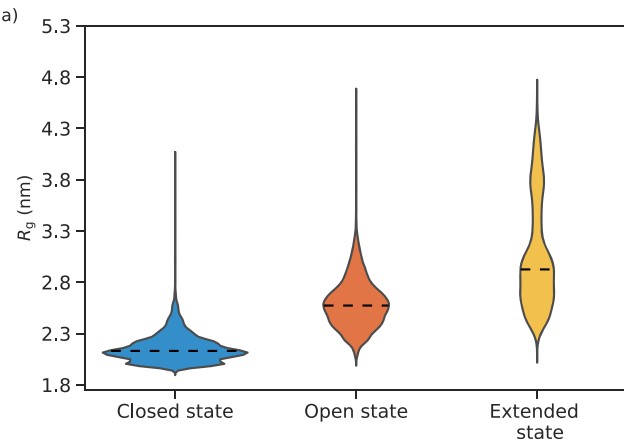

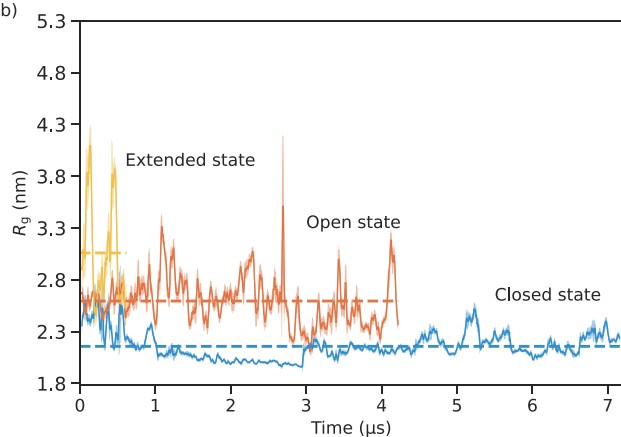

**Fig. 1 | Radius of gyration ($R_g$) of DNAJB6b. a** Violin plots showing the distribution of the radius of gyration ($R_g$) of DNAJB6b in the three observed states. The dashed lines in each violin represents the median of that distribution. For the closed, open, and extended states, the median $R_g$'s are 2.1, 2.6, and 3.1 nm, respectively. The width of each violin corresponds to the number of data points. **b** Variation of $R_g$ with sampling time for the three states and the dashed line indicates the mean $R_g$ for each state. In both (**a**) and (**b**), the $R_g$ data is averaged over 2.5 ns intervals. The spike in the $R_g$ for the open state in (**b**) is due to the conformational change of DNAJB6b just before it transitions into the extended state.

of Phe (F) residues (black dashed lines), i.e., the interactions between the G/F$_2$ and the S/T domain are predominantly $\pi$–$\pi$ in nature (see Supplementary Figs. 4c, d, 5a). In addition, a part of the G/F$_2$ domain (residues 118–125) also interacts with the J-domain through anchor region 2 (AR2), and this is also found in all three states of DNAJB6b (Fig. 2a–c, g–i) implying that AR2, like AR1, is a stable connection that, to the best of our knowledge, has not been reported before. In the closed state, the predominant interactions of the AR2 are cation–$\pi$, and the contacts in the open and the extended states are predominantly aliphatic-aromatic hydrophobic in nature (Fig. 4e–g, k).

In addition, we also found the intradomain contacts within the S/T domain to be predominantly F–F in nature (Supplementary Fig. 4a, b). This leaves the serine (S) and threonine (T) residues of the S/T domain available for intermolecular interactions with substrates such as polyQ[9], and Aβ[11].

### Conformational dynamics of the cycling of DNAJB6b between the closed, open, and extended state

To investigate the molecular origin of DNAJB6b transitioning between the closed, open, and extended states, we calculated the time variation of the number of interdomain contacts. In three out of five simulations, the transition from closed to open relied on the competition between the S/T domain and the CTD for interactions with the G/F$_1$ domain (Supplementary Figs. 6, 7), and in the other two simulations, we observed that the fluctuations in the CTD conformations alone were enough for the CTD to be released from the G/F$_1$ domain thus transitioning to the open state. During this transition, the CTD detaches from both the G/F$_1$ domain and the S/T domain, resulting in a fully free CTD (Fig. 2h). Next, to understand the transition from the open to the extended state, we analyzed the temporal variation in the number of F–F interactions between the G/F$_2$ and the S/T domain and within the S/T domain (Supplementary Fig. 8). As soon as the number of inter- and intradomain F–F interactions reduce, the S/T domain and the CTD move away from the G/F$_2$ domain allowing DNAJB6b to transition into the extended state. No direct transition from the closed to the extended state was observed. In the extended conformation, the S/T domain is predominantly disordered and entirely exposed to the solvent (Fig. 2i and Supplementary Fig. 9b).

Once in the extended state, we observed DNAJB6b to move to a transient, intermediate conformation where the S/T domain interacts with helix I of the J-domain (Supplementary Fig. 10b), after which it transitioned to a near-closed state, that can be interpreted as an on-

pathway transition to the closed state (see Supplementary Fig. 10c). Two additional simulations, using the extended state as an initial configuration, showed that the protein could either convert to the intermediate conformation and further back to the near-open state, suggesting an on-pathway transition to the open state (Supplementary Fig. 10d) or dwelled between the extended state and intermediate conformation. These data show that the extended state is a relatively stable conformation that can transition back to the closed or open states, in both cases, through an intermediate conformation (Supplementary Fig. 10).

From these results, the following picture emerges (Fig. 3): The S/T domain interactions with the G/F$_1$ domain are a trigger for DNAJB6b to transition from the closed to the open state. Only from this open state and upon the loss of F–F contacts within the S/T domain or between the S/T and G/F$_2$ domain, it can transition to the extended state. Once in the extended state, it can either cycle back to the closed or the open state, in both cases by temporarily visiting an intermediate conformation. Hence, based on six independent simulations, combined with the observation that these states were also found in the homology models of Robetta and TR-Rosetta, we conclude that monomeric DNAJB6b converges to a dynamic ensemble of three interconverting states, cyclically changing between closed, open, and extended.

### DNAJB6b: autoinhibition via anchor region 1

Previous studies have shown that Hsp70 interactions with class B JDPs depend on autoregulatory features that are not found in class A JDPs[27]. Autoinhibition is mediated by helix V of the G/F$_1$ domain that interacts with helices II/III of the J-domain, acting as a mask that prevents Hsp70 from accessing the HPD motif, a highly conserved feature of JDPs and known as an essential binding site between E. coli Hsp70 DnaK and DnaJ[40], (Supplementary Fig. 11a). For DNAJB6b, this autoinhibited state was also suggested to be present[24], but this was based on NMR data of a ΔS/T DNAJB6b construct (to ensure it remained monomeric).

First, we verified the existence of autoinhibition for ΔS/T DNAJB6b. Contact analysis indeed revealed the existence of autoinhibition consistent with the NMR findings of Karamanos et al.[24] (see Fig. 4d). Although we observed some AR2 interactions, these were not as prominent as in the full-length DNAJB6b (Fig. 4h). Furthermore, transient interdomain interactions between the J-domain and the CTD, which were not prominent in the full-length DNAJB6b, were observed (Supplementary Fig. 14). In the absence of the S/T domain, the CTD predominantly interacts with the J-domain, whereas in the full-length

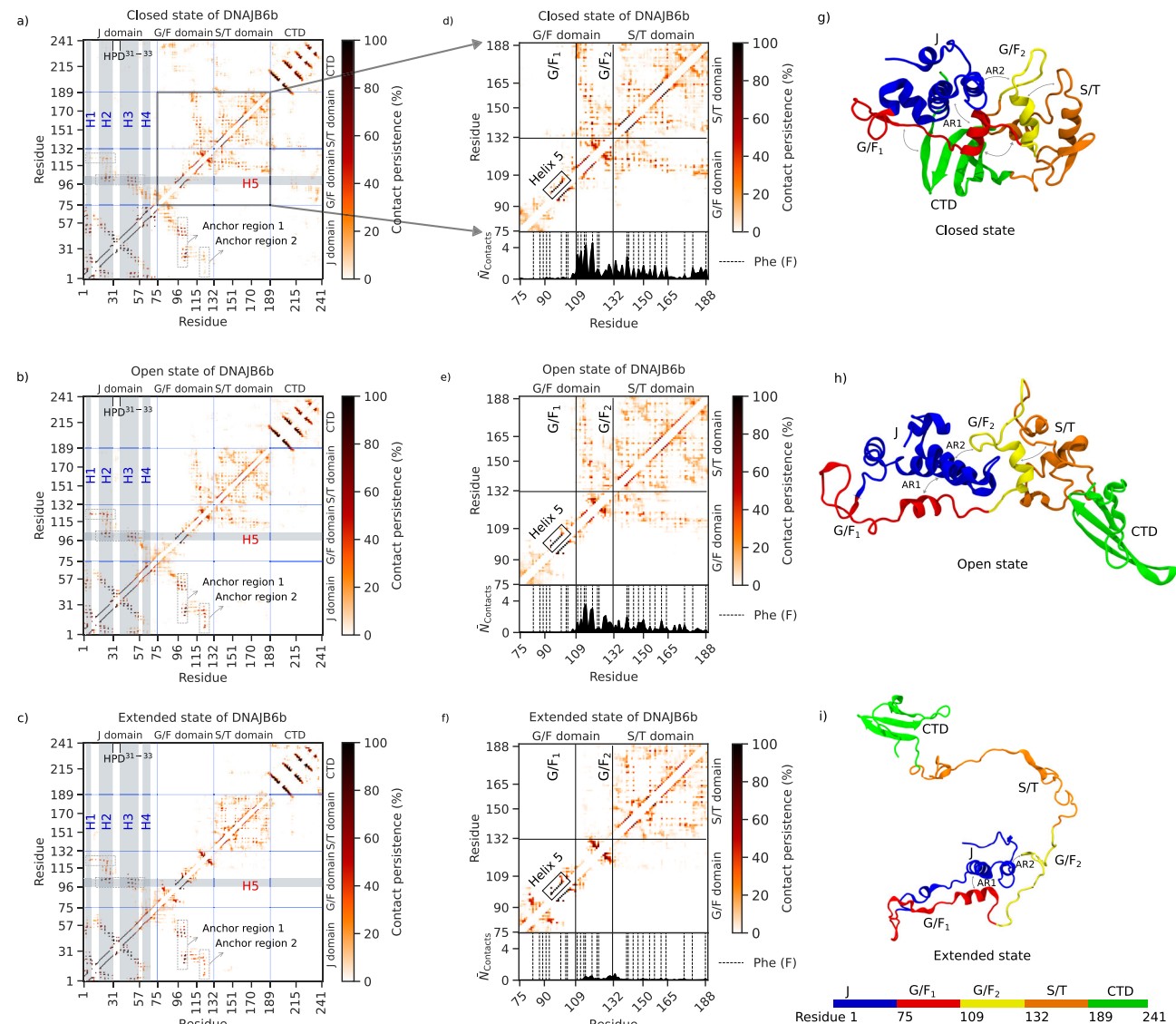

**Fig. 2 | Contact maps and snapshots of the three different states of DNAJB6b.** **a**–**c** Contact maps of the closed, open, and extended states of DNAJB6b. The color bar corresponds to the percentage of time the contact existed during the simulation. The contact map is divided into four sections according to the domains in DNAJB6b, and the division is shown by sky-blue horizontal and vertical lines. The J-domain contains four helices shown by vertical gray bars. The horizontal gray bar shows helix V of the G/F₁ domain. The contacts enclosed in the dashed rectangles indicate the anchor regions 1 and 2 (AR1 and AR2). **d**–**f** Contact maps of the G/F + S/T domains for the closed, open and extended state of DNAJB6b. The 1D contact distribution below the 2D contact map shows the average number of G/F to S/T interdomain contacts. The vertical dashed lines in the 1D contact distribution are the location of the Phe (F) residues. The horizontal and vertical lines separate the G/F domain (G/F₁ + G/F₂) and the S/T domain. **g**–**i** Snapshots showing the molecular structure of DNAJB6b in the closed, open, and extended state. Two-headed arrows summarize the interactions between the domains as observed in the contact maps (**a**–**c**).

DNAJB6b, the CTD interacts with the G/F₁ domain (in the closed state) or does not participate in any of the interdomain interactions (in the open and extended states). From these observations, we conclude that the structural integrity of DNAJB6b is lost upon removing the S/T domain consistent with previous observations[9,11].

Importantly, our data also shows the existence of the auto-inhibitory state for the full-length DNAJB6b. In fact, we observed helix V to be strongly interacting with helices II/III in all three states (closed, open, and extended) (AR1 in Figs. 2, 4a–c). The nature of the molecular interactions involved in AR1 are predominantly hydrophobic (mostly aliphatic-aromatic but also aromatic, π–π), followed by cation–π and electrostatic (Fig. 4j). Upon a closer look at the autoinhibition, we found that for all three states, residue F100 of the conserved DVF motif[29,30] of helix V extends towards helices II/III and is situated in the center, participating in numerous interactions (aliphatic and aromatic hydrophobic, see Fig. 4i), making it a key residue for the overall

stability of the autoinhibitory state (AR1). Furthermore, we noticed in all three states that, H31 of the HPD motif, crucial for the ATPase activity of Hsp70, interacts with F104 of helix V, making H31 occupied and unexposed to the solvent. These findings (that autoinhibition is never lost and H31 is always buried despite the dynamic behavior of DNAJB6b) create an intriguing possibility that autoinhibition may prevent the binding of Hsp70 in a substrate-free state.

**Spontaneous loss of autoinhibition in DNAJB6b is a feature of LGMDD1-associated mutations**

Several mutations in DNAJB6b have been shown to cause LGMDD1, a muscle degenerative disease associated with the accumulation of protein aggregates. Most mutations are actually found in the G/F₁ region of the protein[17,18,41,42], but the precise structural and functional effect of these mutants is yet poorly understood. To investigate the effect of LGMD mutations on the tertiary structure and autoinhibitory

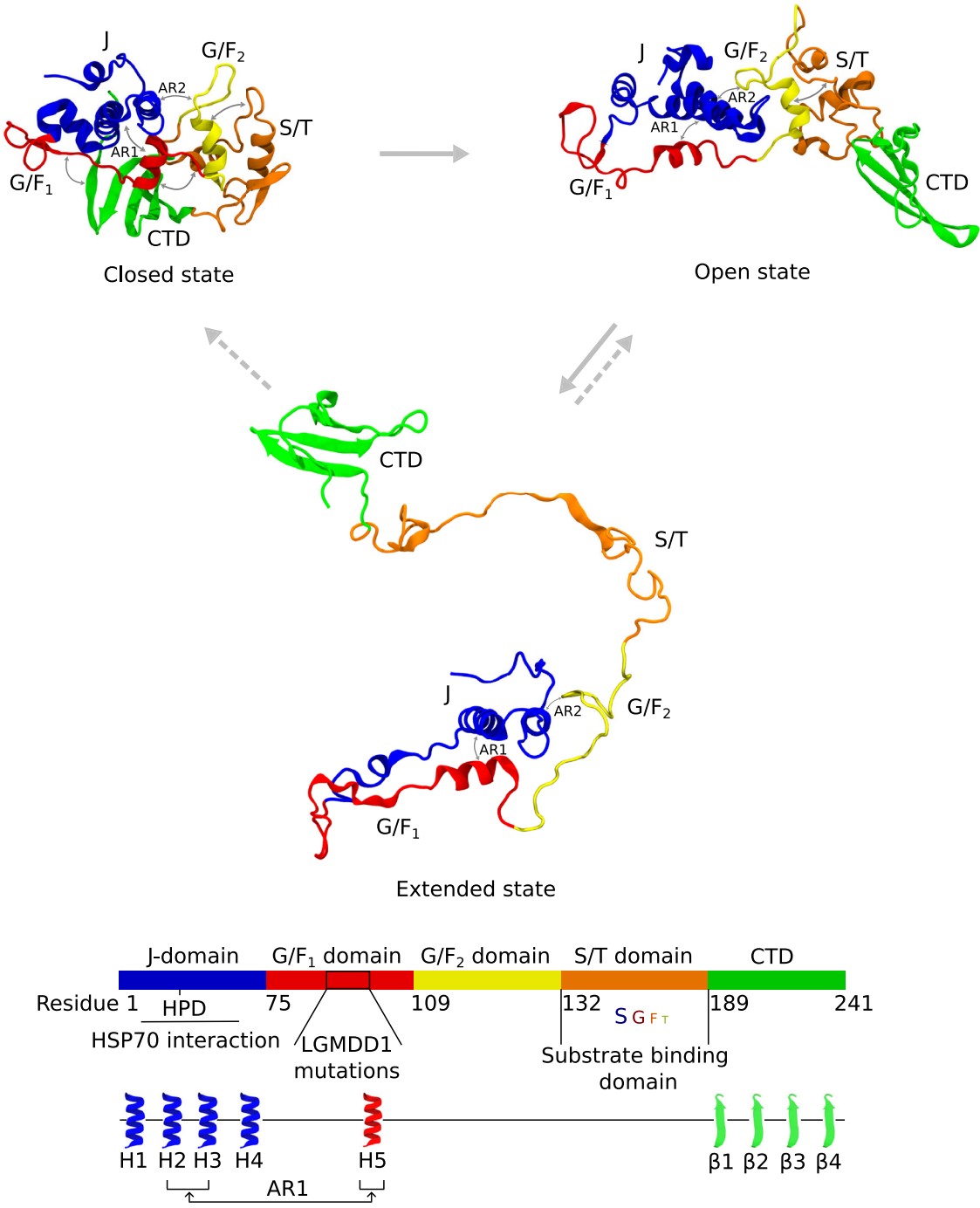

**Fig. 3 | DNAJB6b transiently converts between the three different states.** Figure depicting the transitions between the open, closed, and extended states of DNAJB6b. Two-headed arrows summarize the interactions between the domains as observed in the contact maps (see Fig. 2). The distinction between the open, closed, and extended states of DNAJB6b is made on the basis of the number of contacts between the different regions (see "Analysis of the MD simulations" in the Methods section). In all three states, anchor regions 1 and 2 are present. Anchor region 1 (AR1) consists of interactions between helix V of the G/F₁ domain and helices II/III of the J-domain, forming the autoinhibitory state of DNAJB6b. Anchor region 2 (AR2) denotes the interactions between residues 118–125 of the G/F₂ domain and residues 10–30 of the J-domain. These are the only interdomain contacts in the extended state. In the open state, we have additional contacts between the G/F₂ and the S/T domain, and in the closed state, we have additional contacts of the CTD with the G/F₁ and S/T domains. The straight gray arrows denote the interconversions between the three states, and the dashed gray arrows denote the transitions from the extended state to the near-closed and open states.

state of DNAJB6b, we performed five distinct point mutated DNAJB6b simulations (F91L, F93L, P96R, F100I/V), see Methods section for details.

In all the simulations performed, we observed an increase in fluctuations of the residues in the G/F₁ linker related to a loss of interaction with helix IV of the J-domain (Fig. 5a and Supplementary Fig. 13). Additionally, we observed a slight decrease in the number of contacts between helix V and helices II/III, responsible for the auto-inhibitory state (Supplementary Fig. 12). For the key residue F100 of helix V, we noticed a loss of interaction with helices II/III for the mutations F100I or F100V. However, within the simulated time scale, complete destabilization of the autoinhibition was not observed, likely

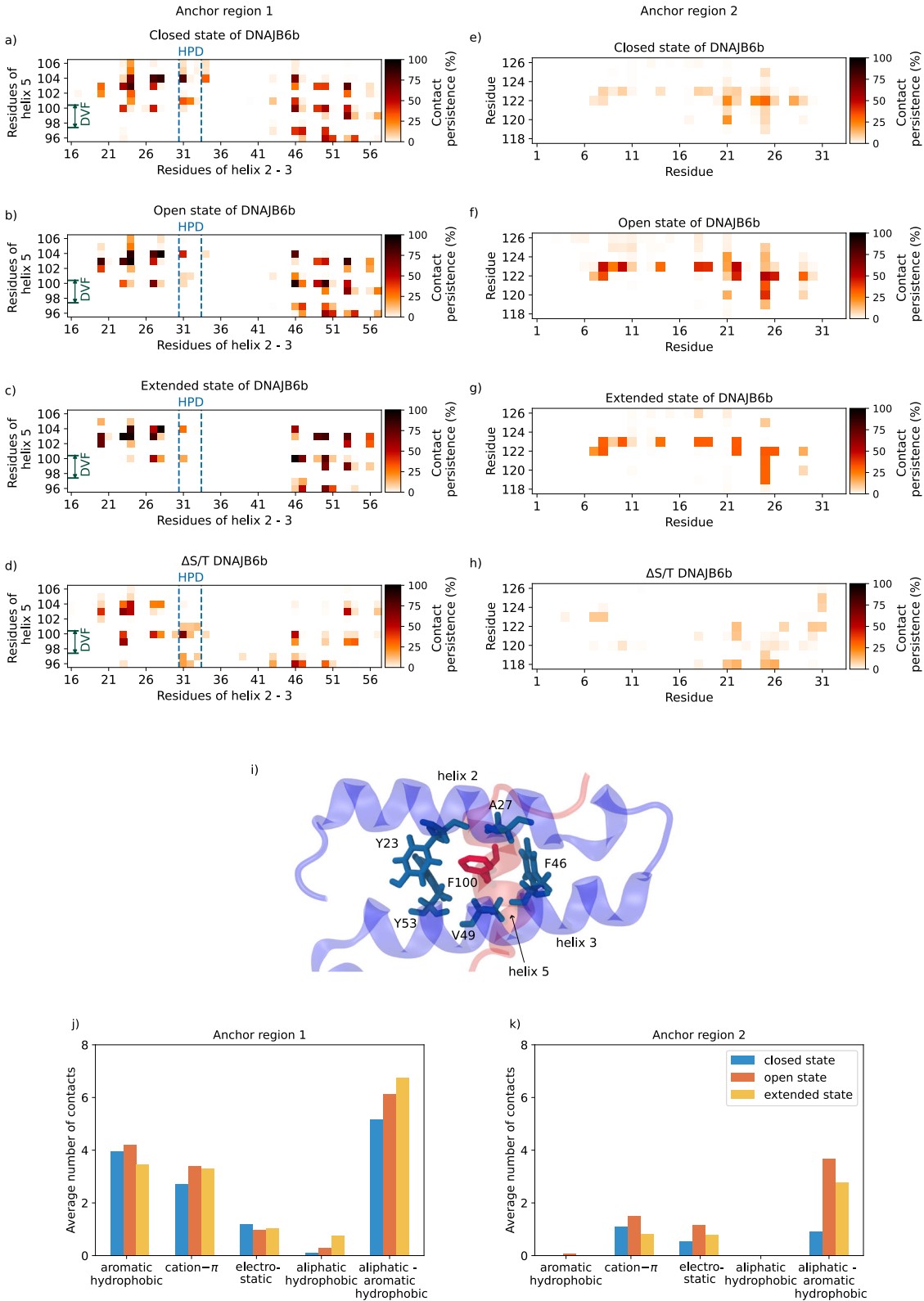

**Fig. 4 | Contact maps of anchor regions 1 and 2 of DNAJB6b. a–c** Contact map of anchor region 1, i.e., interactions between helices II/III (residues 16 to 56, including the loop connecting them) of the J-domain and helix V (residues 96 to 104) of the G/F₁ domain in the closed, open and extended states of full-length DNAJB6b and **d** ΔS/T DNAJB6b. The blue dotted lines indicate the HPD motif (residues 31–33). It can be seen that residue H31 forms a key interaction with residue F104. (caption continued on next page) **e–h** Contact map showing anchor region 2, i.e., interactions between residues 118 to 125 of the G/F₂ domain and the N-terminal residues of the J-domain.

The color bar represents the percentage of time the contact existed during the simulation. It can be noted that anchor region 2 for ΔS/T DNAJB6b is almost completely absent. **i** Snapshot showing F100 of helix V extending towards helices II/III of the J-domain and participating in multiple interactions with aliphatic and aromatic hydrophobic residues. Bar plot showing the average number of contacts of a specific type observed in figures (**a–c**), (**e–g**) for **j** anchor region 1 (AR1) and **k** anchor region 2 (AR2), respectively. See Methods for details.

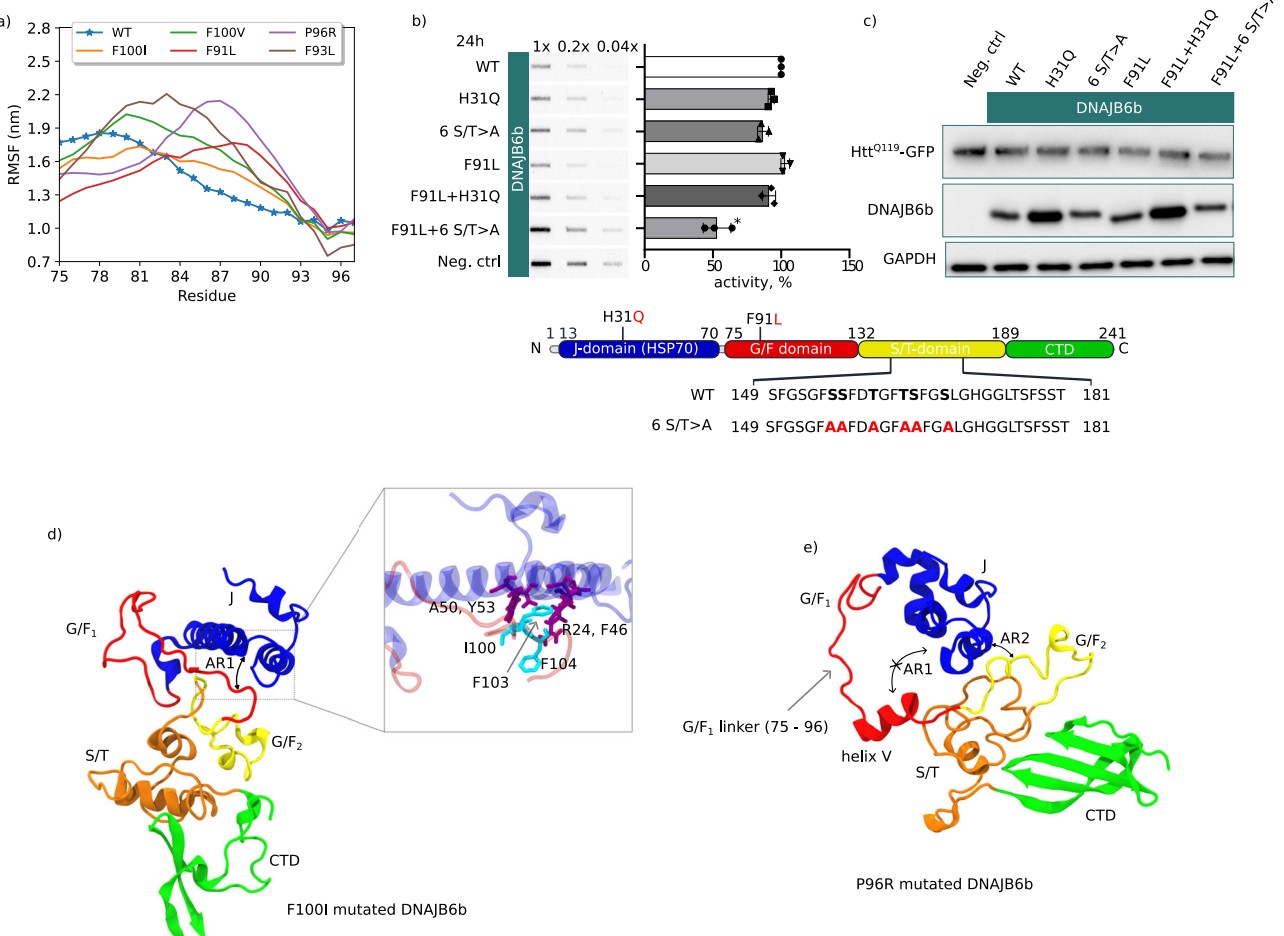

**Fig. 5 | Effect of LGMDD1 mutations on the autoinhibitive state of DNAJB6b.**
**a** Root mean square fluctuations (RMSF) of the G/F$_1$ linker for the open state of WT-DNAJB6b and the LGMDD1-associated point mutations. It can be noticed that the fluctuations in the G/F$_1$ linker are higher for the LGMDD1-mutated DNAJB6b compared to the WT-DNAJB6b. **b** Htt$^{Q119}$-GFP aggregation in the presence of different DNAJB6b variants (filter trap assay, incubation time 24h). $N = 3$ independent biological replicas were performed. To compare WT and F91L + 6 S/T > A, an unpaired $t$-test with Welch's correction, two-tailed, $P = 0.0153$ is used. Data were presented as mean values ± SD. **c** Expression levels of Htt$^{Q119}$-GFP and DNAJB6b (Western blot).

Western blot has been done once to ensure that expression of the mutant versions of DNAJB6b is comparable to the wild-type version, and to test if the soluble levels of Q119 were similar between the samples. **d** Snapshot showing the partial destabilization of anchor region 1 (AR1) for the F100I mutated DNAJB6b. It can be seen that upon mutation, I100 lost its interaction with the J-domain, but the loss of interactions was partially compensated by F103 and F104 (inset). **e** Snapshot showing the complete destabilization of AR1 for the P96R mutation due to extensive fluctuations in the G/F$_1$ linker (Fig. 5a). Despite the AR1 destabilization, anchor region 2 (AR2) remained stable.

because residues F103 and F104 partially compensated for F100 in the interaction with the J-domain (Fig. 5d).

In one of the P96R mutation simulations, we did observe a complete destabilization of the autoinhibitory state at around ~1700 ns which remained destabilized for the duration of the simulation (an additional 300 ns). In this mutant, residue R96 engaged in a cation−π interaction with F100, thus inhibiting F100 from interacting with the J-domain. This decrease in interdomain interaction combined with significant fluctuations of the G/F$_1$ linker region ultimately led helix V to move out completely from its original position, thus exposing the binding sites on helices II/III (Fig. 5e). The destabilization of DNAJB6b was observed during its open state but not in the closed state. As mentioned earlier, the CTD interacts with the G/F$_1$ domain in the closed state, and we hypothesize that this additional interaction serves a protective role, preventing the displacement of helix V from its original position and maintaining the stability of anchor region 1.

The destabilization of autoinhibition for the P96R mutant, together with the induced fluctuations of the residues in the G/F$_1$ linker for all the LGMD mutants, suggests that LGMD-causing mutations in the G/F$_1$ domain may have an impact on the stability of the autoinhibitory state potentially leading to the release of helix V from helices II/III and

hence allowing access of Hsp70 in the absence of substrates, in line with the findings of Abayev-Avraham et al. in the accompanying paper[32].

To extrapolate this to functionality, we turned to cellular experiments and addressed how LGMDD1 mutants of DNAJB6b might affect its anti-aggregation properties towards polyglutamine proteins. It was shown before[6,18] that some LGMDD1 mutants show a minor loss of function (LOF) in preventing polyQ aggregation compared to wild-type DNAJB6b. However, we did not detect a functional defect in the DNAJB6b-F91L mutant (Fig. 5b, c), which might either be due to different severity of the F91L mutant or to the experimental setup of the present study. However, it supports the notion that the G/F$_1$ region (with LGMD mutants) is not critical for the 'holdase' function required primarily in preventing polyQ aggregation in cells[7,9]. It is also in line with recent in vitro data showing that LGMD mutants are not affecting the substrate binding and their 'holdase' capacity to suppress aggregation of polyQ and TDP43[32]. In fact, the same was observed in cells for the DNAJB6b-H/Q mutant that has a fully intact substrate binding domain, but that cannot interact with Hsp70[7,9]. It supports the notion that the G/F$_1$ region (with LGMD mutants) is not involved in preventing polyQ aggregation in cells[7,9], and it is in line with recent in vitro data

showing that LGMD mutants are not affecting substrate binding and chaperone capacity[32]. Interaction of DNAJB6b with Hsp70 becomes only important after substrate loading for substrate processing[7,9]. The impact of this H/Q mutation, i.e., improper interaction with Hsp70, is only revealed when combined with a second mutant that alone slightly impairs this substrate binding and anti-polyQ aggregation capacity due to a replacement of five of the serine (S) and threonines (T) of the substrate binding domain into alanines (6 S/T > A). This double mutant now shows a complete loss of function[9]. When combining the F91L mutant with such a second substrate binding impairing mutation, we also find more than additive effects, and the double mutant (H/Q + F91L) is strongly impaired in preventing polyQ aggregation (Fig. 5b). In contrast, introducing an H/Q mutation in the background of F91L mutant show no additive effects (Fig. 5b) as they both affect Hsp70 interactions.

## Discussion

J-domain proteins play a key role in the regulation and functional specification of the Hsp70 chaperone machines in cells. Based on domain architectural criteria, the many JDPs have been categorized into class A, B, and C types[5]. DNAJB6b, which we investigated here, belongs to the class B type. Yet, despite some similarities, the class B members have distinct functional characteristics and show several structural differences. Evolutionary analysis actually divides this class into two subclasses: the DNAJB1-like proteins and the DNAJB6-like proteins[43]. Using homology modeling and all-atom molecular dynamics simulations, we show that the monomer DNAJB6b is an intrinsically highly dynamic protein that cycles between three states: a closed state, an open state (both abundant), and a less abundant extended state that only forms from the open state. Interestingly, these three states were also found among the multiple states predicted by homology modeling: the highly extended conformation of the S/T domain was predicted by AlphaFold and the models generated by Robetta and TR-Rosetta predicted a mix of closed (4) and open states (6). Similar to the DNAJB1-like proteins[27], the G/F region contains a helix V that blocks access of Hsp70 to the J-domain (autoinhibition), shown by the NMR data of Karamanos et al. [24] using a DNAJB6b construct lacking the S/T domain. Our simulations not only recapitulate these findings for this ΔS/T DNAJB6b construct but also reveal that this autoinhibition is present in the full-length protein as well and, in fact, is stable in all three different states we detected. This is consistent with findings on DNAJB8, the closest homolog of DNAJB6b[28], but in apparent contrast with the findings of Soderberg et al. [31] using homology modeling and structural analysis on full-length oligomeric DNAJB6b.

This urges the question of what could drive the loss of these autoinhibitory interactions. For DNAJB8, it had been hypothesized that the release of the CTD interaction with JD and GF might drive the opening of helix II/III for Hsp70 access[28]. While our models show that also for DNAJB6b, the CTD is in close proximity to helices II/III, which could physically obstruct Hsp70 access, this is only true for the closed state. In both the open and extended state, our model predicts that helices II/III remain obstructed by helix V, without the requirement of CTD to be in close proximity. Clearly, when analyzed in isolation, i.e., without substrate or Hsp70 being present, the autoinhibitory state of the monomeric DNAJB6b is not released, irrespective of the three individual states. Since the conversion from the closed to open state is dependent on interactions between the G/F$_1$ region and the 4-β-strand containing CTD, our data suggest that the loss of the CTD-G/F$_1$ interaction is necessary, but not sufficient to release autoinhibition. For DNAJB1 relieving autoinhibition was shown to be mediated by Hsp70 itself through its C-terminal EEVD motif that binds to the N-terminal domain of DNAJB1[27]. This is supported by findings that mutations in or deletions of the EEVD motif of Hsp70s impair their ability to bind to class B J-proteins and functionally stimulate their chaperone activity[44–46], while they still can do this with class A J-proteins[45,47] that

seem to lack this autoinhibitory mechanism[27]. It has been indicated for DNAJB6b's ortholog DNAJB8 that the EEVD tail does not bind[48], which is also shown to be valid for DNAJB6b in the co-submitted paper by the group of Rosenzweig[32]. Our simulations do not exclude the possibility of an active (non-EEVD-like) competition between Hsp70 and helix V for binding. However, our data favor an alternative and functionally highly intriguing mechanism in which relief of autoinhibition only occurs upon and is mediated by allosteric effects of substrate binding to the S/T region as was also proposed for DNAJB8[28]. This mechanism would functionally couple DNAJB6b-substrate interaction to Hsp70 recruitment for further processing. However, the molecular details on how this mechanism can lead to compaction of the S/T-substrate complex in order to transfer the substrate to HSP70 at the J-domain is unknown at this stage.

The importance of autoinhibition for function is supported by the findings of the LGMD mutants. Cellular data have shown that, under certain conditions, DNAJB6b LGMD mutations have dominant negative effects on chaperone functions that can be negated by abrogating their interactions with Hsp70 by mutating the HPD motif[49]. Under other conditions and for other substrates, the same LGMD mutants also show a loss of functionality[6,18], and we show here that this is related to Hsp70 interaction rather than to substrate interaction per se (Fig. 5b). Our MD data suggest that these LGMD mutants can drive the spontaneous release of autoinhibition. Within the time frame of the MD simulations, we observed one complete destabilization event for the P96R mutant. Furthermore, we show that the G/F$_1$ linker region of the LGMD mutated DNAJB6b is more dynamic than that of the wild-type DNAJB6b, and this is associated with an increased propensity of the loss of contacts between helix V and helices II/III over time, which might lead to increased access of Hsp70 to DNAJB6b without substrates bound. In line, the co-submitted work of the group of Rosenzweig[32] using NMR and biochemical analysis shows that the LGMD pathogenic mutants are unaffected in substrate interactions and chaperoning but rather lead to unregulated binding and hyperactivation of the ATPase of Hsp70 in the absence of clients. Combined, these findings suggest a spontaneous release of autoinhibition upon LGMD mutations. An important question that remains is why LGMD mutants under some conditions seem to have a dominant negative effect[19,49] whereas in other situations, they only seem to be associated with (minor) loss of function[6,18]. A possible explanation might be related to the different expression levels of Hsp70 under different stress conditions. Basal expression levels of Hsp70 have been shown to be not rate limiting for substrates like polyQ proteins as evidenced by near to no beneficial effects of overexpression of Hsp70 and by the fact that DNAJB6b lacking the HPD motif (and hence cannot interact with Hsp70) is quite effective in suppressing polyQ aggregation[6,18]. On the other hand, under acute stress conditions like heat shock (as used to asses TDP43 handling by LGMD DNAJB6b mutants[19]), levels of Hsp70 rapidly become rate limiting and are strictly correlated with cellular resilience and protein aggregate handling[50]. The latter situation could be more reflective of the situation in muscles which are under constant (stretch-related) stress, which would hence also explain why over-expression of LGMD mutants in these tissues causes degenerative features[49,51]. However, inhibition of DNAJB6b or its interactions with Hsp70 should be considered with caution, given that it was recently found that depletion of DNAJB6b leads to problems with nuclear pore assembly and causes a decline in nucleocytoplasmic shuttling[16].

We report the existence of an extended state in which the S/T domain is almost fully disordered, maximally exposing the S, T, and F residues for substrate binding and oligomerization. The inter-molecular interactions that seem crucial for the transition of DNAJB6b from the open to this extended state are those between the G/F$_2$ region and the S/T region. These interactions are predominantly mediated by F−F (π−π) binding and the release of these contacts is the key event that triggers the conversion into the extended state. The work of

Kuiper et al.[16], highlighted the critical role of the phenylalanine (F) residues in the S/T domain in the ability of DNAJB6 to suppress the aggregation of FG-Nup fragments. In our simulations, these phenylalanine (F) residues predominantly interact with the G/F$_2$ domain in both the open and closed states (Fig. 2d, e), and are only exposed in the extended state (Fig. 2f). Since the serine (S) and threonine (T) residues of the S/T region are crucial for DNAJB6b to chaperone polyQ[9] and Aβ[11] and the phenylalanine (F) residues of the S/T domain are crucial for preventing the aggregation of FG-Nup fragments[16], the enhanced exposure of the S, T, and F residues renders the extended state to be highly efficient compared to both open and closed states. Hence, it will be worth studying if and how the interaction between the G/F$_2$ and the S/T region may be regulated to stimulate DNAJB6b function.

Another still open question relates to the role of self-oligomerization of the DNAJB6-subgroup. A recent study on DNAJB8 revealed that this is related to a specific peptide ($^{147}$AFSSFN$^{152}$) within the S/T-rich region[52]. It was found that for DNAJB8, mutations in this peptide did abolish self-oligomerization and lead to DNAJB8 monomerization; this monomeric DNAJB8, however, did retain wild-type activity in terms of substrate binding and chaperone function[52]. We speculate that for DNAJB6b, substrate binding and self-oligomerization could be driven by the same S/T domain of DNAJB6b, and that monomers are the functional chaperone moieties, and self-oligomerization may only happen at high concentrations in the absence of clients. Whether such substrate-free oligomers exist in cells remains to be established, but it is tempting to speculate that these could act as reservoirs of chaperone activity under conditions of acute needs, as has, for instance, also been suggested for oligomeric assemblies of small Hsp[53,54]. For small Hsp, de-oligomerization has been suggested to be regulated by (stress-induced) post-translational modifications like phosphorylation[55,56]. How this would work for DNAJB6-like proteins is yet unknown, but it could even involve Hsp70-dependent disassembly as the J-domains seem accessible in the oligomeric form (Soderberg et al. [31]).

In summary, our molecular dynamics data suggest that DNAJB6b monomers are not only intrinsically highly dynamic proteins but, at the same time, are also more structured than predicted by e.g., alphafold, with several intramolecular interactions that point to a high degree of functional regulation both for substrate interactions as well as for access to Hsp70 for substrate processing. The importance of this regulation is highlighted by the finding that Hsp70 access is restricted in all non-substrate loaded DNAJB6b monomers (our MD data) and that mutants that cause LGMD relieve this restriction (Abayev-Avraham et al. [32], our MD data).

## Methods

### Molecular dynamics simulations

All-atom explicit-water molecular dynamics (MD) simulations were performed using the GROMACS MD package, version 2021.3[57]. The CHARMM36m forcefield with TIP3P water and the amberff99SBdisp with a99SB-disp water is used to study the dynamics of DNAJB6b[58,59]. Three different replicas were performed for each forcefield for statistical rigor, adding up to six different simulations (Supplementary Fig. 2). A leap-frog algorithm is used for integrating Newton's equation of motion with a timestep of 1 fs. All the simulations were performed for at least 2 µs each, totaling 12 µs of simulation time. For CHARMM36m, a cutoff of 1.2 nm is used for short-range electrostatics and van der Waals interactions, whereas in amberff99SBdisp, a cutoff of 1.0 nm is used. A long-range dispersion correction for energy and pressure was applied for the case of the amber forcefield[59]. In both the forcefields, the long-range interactions are evaluated using the particle mesh Ewald method with a grid spacing of 0.125 nm. A modified Berendsen thermostat[60] with a time constant of 0.1 ps is used to maintain the temperature of the system at 300 K. For simulations with the NPT ensemble, the Parinello–Rahman barostat[61] is used with a time

constant of 1 ps to maintain a pressure of 1 bar. Bonds involving hydrogen atoms were constrained using the LINCS algorithm[62].

DNAJB6b is placed at the center of a cubic box and solvated with water with a padding of 1.5 nm. Na$^+$ and Cl$^-$ ions were added to the system by replacing water molecules to obtain a salt concentration of 100 mM. The resulting system is then subjected to energy minimization using the steepest descent algorithm. To equilibrate the solvent and the ions around the protein, we have performed two consecutive equilibration simulations using the NVT (for the temperature to reach a plateau) and NPT ensemble (for the density to reach a plateau) for 1 ns each. Finally, the production runs are performed in the NPT ensemble for at least 2 µs.

For the additional simulations starting from the extended state (Supplementary Fig. 10), we obtained the tertiary structure at 1297 ns from simulation A2 (Supplementary Fig. 2). The obtained tertiary structure is used as an initial configuration, and two simulations were performed with the two different forcefields. The simulation protocol we followed for these simulations is similar to the regular DNAJB6b simulations described above. For the LGMDD1 mutant simulations, mutations on the Robetta model #4 were incorporated using VMD, and the resulting structure is used as an initial configuration for the simulations. We incorporated F91L, F93L, P96R, and F100I/V point mutations on DNAJB6b, and for each point mutation, we performed two replicas using the two forcefields, for 2 µs each, totaling 40 µs of simulation time. A similar simulation protocol as the wild-type DNAJB6b simulations was followed for LGMDD1 mutation simulations, which includes steps such as energy minimization, equilibration, and production.

For the ΔS/T DNAJB6b simulations, the NMR structure (PDB ID 6U3R) is used as the initial configuration for the simulations. Simulations were performed using three forcefields CHARMM36m, amberff99SBdisp, and CHARMM36–WYF[58,59,63]. Three replicas of simulations were performed for each forcefield for 2 µs each. A similar simulation protocol as the full-length DNAJB6b simulations were followed for the ΔS/T DNAJB6b simulations, which includes steps such as energy minimization, equilibration, and production.

### Analysis of the MD simulations

A contact analysis is performed for all the simulations and plotted as a heat map within a percentage range of [0,100]. A contact is defined if an atom of residue X comes into a cutoff distance of any atom of residue Y (X ≠ Y). The cutoff distance used in this study is 4 Å. Hydrogen atoms are excluded while doing the contact analysis. In addition, contacts between residue $i$ and $i + 1$ until $i + 3$ were excluded during the contact analysis. We used contacts as a criterion to differentiate between the states of DNAJB6b. If a contact exists between the G/F$_1$ domain and the CTD, then DNAJB6b is considered closed; otherwise, it is in an open or extended state. We imposed two conditions for considering the DNAJB6b to in the extended state: 1. The CTD should not interact with the G/F domain, 2. The number of contacts between the G/F$_2$ domain and the S/T domain is less than four. For the analysis all simulation trajectories were initially concatenated using GROMAC's inbuilt tool `trjconv`. Subsequently, this concatenated trajectory was partitioned into three separate trajectories corresponding to the closed, open, and extended states.

To quantify the approximate size of the protein, the radius of gyration ($R_g$) is calculated using GROMACS's inbuilt analysis tool `gyrate`. Secondary structure calculations are performed using the STRIDE algorithm[64], where coils and turns are considered disordered, and the rest (helix, β-sheet, etc.) is considered ordered structure.

Root mean square fluctuations (RMSF) are calculated to measure the fluctuations of a residue relative to a reference. Prior to the RMSF calculation, the rotational and translational motion of the protein are removed using the GROMACS inbuilt tool `trjconv`. The alignment is done using the VMDs inbuilt `measure fit` command with the protein's

backbone as the selection with the average structure of the open state of WT-DNAJB6b as the reference. The RMSF of the backbone is computed for the entire trajectory using the `measure rmsf` command.

We categorized inter-residue interactions into five types. Aromatic hydrophobic interactions encompass contacts among aromatic residues, including phenylalanine (F), tyrosine (Y), and tryptophan (W). Cation-$\pi$ interactions are recognized as contacts between a positively charged residue (arginine (R), lysine (K), or histidine (H)) and an aromatic residue (F, Y, or W). Electrostatic interactions involve contacts between positively charged residues (R, K, or H) and negatively charged residues like aspartate (D) or glutamate (E). Aliphatic hydrophobic interactions include contacts between aliphatic residues such as glycine (G), alanine (A), valine (V), leucine (L), isoleucine (I), and methionine (M). Aliphatic-aromatic hydrophobic interactions are identified as contacts between aromatic residues (F, Y, W) and aliphatic residues (G, A, V, L, I, M).

Calculation of the average number of contacts in the bar plot for Fig. 4j are done for anchor region 1, i.e., between residues 16 to 56 of the J-domain and residues 96 to 104 (helix V) of the G/$F_1$ domain. For anchor region 2, in Fig. 4k, the average number of contacts are calculated between residues 118 to 125 of the G/$F_2$ domain and the residues 1 to 28 (helices II/III) of the J-domain.

### Cell culture and transient transfection

HEK293T cells (not authenticated) were cultured at 37° and 5% in DMEM (Gibco) supplemented with 10% FBS (Fetal bovine serum, Bodinco) and 100 U/mL penicillin/streptomycin (Invitrogen). For transient transfections, $5 \times 10^5$ cells were plated on 35-mm cell culture dishes (Nunc) 1 day before. Cells were transfected with 2 µg of plasmid (0.2 µg pQ119-GFP, 1.8 µg pDNAJB8 variant/empty vector) using polyethyleneimine (PEI) in 1:3 ratio.

### Protein extraction for FTA and western blot

Twenty-four hours after transfection, cells were washed twice with cold phosphate-buffer saline (PBS), scraped in 230 µL of lysis buffer (50 mM TRIS-HCl pH 7.4, 100 mM NaCl, 1 mM MgCl₂, 0.5% SDS, EDTA free complete protease inhibitors cocktail (Roche), and 50 units/mL Denarase (c-LEcta)), and left on ice for 30 min with occasional vortexing. After incubation, the SDS concentration was increased to 2%. Protein concentrations were measured with a DC protein assay (Bio-Rad), equalized using dilution buffer (50 mM TRIS-HCl pH 7.4, 100 mM NaCl, 1 mM MgCl₂, 2% SDS, 50 mM DTT), boiled for 5 min, and stored at −20 °C.

### Filter trap assay

For filter trap assay (FTA), prepared samples were diluted 5-fold and 25-fold in dilution buffer. Then, both original (1x) and diluted (0.2x, 0.04x) samples were applied onto a 0.2-µm pore Cellulose acetate membrane, prewashed with FTA buffer (10 mM TRIS-HCl pH 8.0, 150 mM NaCl, 0.1% SDS). The membrane was washed under mild suction three times with FTA buffer, blocked in 10% non-fat milk, and blotted with anti-GFP/YFP antibody (mouse monoclonal IgG2, JL-8, Clontech, Cat No. 632381) at 1:5000 dilution overnight at 4 °C on a rocking platform. Then the membrane was incubated with an anti-mouse HRP-conjugated secondary antibody (sheep, GE Healthcare, Cat No. NXA931) diluted 1:5000 in 5% non-fat milk and visualized with enhanced chemiluminescence using a ChemiDoc Imaging System (Bio-Rad). Signal intensities were measured in the ImageLab software, and the results of 1x and 0.2x samples (where possible) were averaged. Values were normalized to the control sample, analyzed with Graph-Pad Prism, and plotted in a graph. The resulting graph represents an average of three separate experiments.

### Western blot analysis

For the WB analysis, samples aliquots were mixed with 4x Sample buffer (50 mM Tris-HCl pH 6.7, 2% SDS, 10% glycerol, 12.5 mM EDTA, 0.02% Bromophenol blue). Samples were loaded on 10% SDS-PAGE gel and ran at 90 V. Proteins were transferred to a nitrocellulose membrane (Schleicher and Schuell, PerkinElmer, Waltham, MA, USA), blocked in 10% non-fat milk, and blotted with primary anti-GAPDH antibody (mouse monoclonal clone GAPDH-71.1, Sigma, Cat No. G8795) diluted 1:5000, primary anti-GFP/YFP antibody (mouse monoclonal IgG2, JL-8, Clontech) diluted 1:5000, or with anti-V5 antibody (rabbit polyclonal, Abcam, Cat No. ab9116) diluted 1:5000, overnight at 4 °C on a rocking platform. After that, the membrane was incubated with an appropriate (mouse or rabbit) HRP-conjugated secondary antibody (anti-mouse antibody, sheep, GE Healthcare, Cat No. NXA931; anti-rabbit antibody, donkey, GE Healthcare, Cat No. NA934) diluted 1:5000 in 5% non-fat milk and visualized with enhanced chemiluminescence using a ChemiDoc Imaging System (Bio-Rad).

### Reporting summary

Further information on research design is available in the Nature Portfolio Reporting Summary linked to this article.

## Data availability

All study data are included in the article and/or supplementary text. The NMR Structure of ΔS/T DNAJB6b is taken from PDB ID 6U3R for performing ΔS/T DNAJB6b simulations. The initial configuration files for replicating the MD simulations, and average structures for closed, open, and extended states have been deposited in the Zenodo database https://zenodo.org/records/10592232. The data generated in this study are provided as a source data file and are provided with this paper. Analysis of plotting codes and any additional data related to this paper are available from the corresponding author upon request. Source data are provided with this paper.

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

## Acknowledgements

This work was financially supported by the Netherlands Organization of Scientific Research grant no. OCENW.GROOT.2019.068. This work made use of the Dutch national e-infrastructure with the support of the SURF Cooperative using grant no. EINF-4927. We thank the Center for Information Technology of the University of Groningen for their support and for providing access to the Peregrine and Hábrók high-performance computing cluster. We acknowledge the fruitful discussions with Prof. dr. L.M. Veenhoff.

## Author contributions

P.R.O. and H.H.K. designed the project. Simulations are designed by V.A. and P.R.O. In cellulo experiments are designed by E.U., P.R.O. and H.H.K. V.A. performed the simulations and in cellulo experiments are performed by E.U. Manuscript setup by V.A., P.R.O. and H.H.K. and editing by all.

## Competing interests

The authors declare no competing interests.
