## [Peer Review File · Nature Communications]

Tertiary structure and conformational dynamics of the anti-amyloidogenic chaperone DNAJB6b at atomistic resolutionREVIEWER COMMENTS

Reviewer #1 (Remarks to the Author):

This is an intriguing paper using molecular dynamics simulations to try and understand potential intramolecular interactions of DNAJB6, which have been difficult to study by wet-lab approaches. The simulations raise interesting hypotheses about the ST region and CTD regulating access to the J-domain by Hsp70s, which may be linked to substrate binding. The hypothesis is very convincing in consideration of the biological context (inefficient to bind/sequester an Hsp70 unless there is substrate to deliver). It can be difficult however to be convinced by the specific details of the DNAJB6 structures and their interconversions in the absence of experimental validation (for example from H/D exchange mass spec or perhaps additional cross-linking mass spec). It was generally quite difficult to know which details of the models had high confidence and where the limits of interpretation might lie. At the same time, it is appreciated that it has been very difficult to get atomic resolution information on the native, full-length version of this protein, and the sort of structural framework proposed could be significant in moving the field forward with testable hypotheses.

Confidence that the chosen starting structure ('closed') is relevant in terms of the JD/CTD arrangement, or even whether it is the most relevant from the list of structures in Supp Table 1 is not high considering the information provided. The closed structure is chosen from a set of structures from Robetta, RaptorX, TR-Rosetta, or AlphaFold and based on agreement with CLMS data. Firstly, none the structures is able to satisfy all the lysine crosslinks (in Robetta #4, K196-K29 is 40Å, well above the standard cut-off). In addition, K225 crosslinks K21 and K70 which are on opposite faces of the J-domain, and K196 and K225 (on one face of the CTD beta sheet) crosslink to K29 and K60, respectively, which are on opposite faces of the JD. These crosslinks suggest there is more than one docking pose and/or a high degree of flexibility. Second, the TR-Rosetta #1 structure matches the CLMS data as well as Robetta #4 (20.46 vs. 19.07). How different are TR-Rosetta #1 and Robetta #4 from each other, and are there other criteria that might justify using one over the other?

Related to the above, are there common differences between the entire set of structures that might give insights into potential conformational changes, which might complement the results from the MD simulations?

This is a system with lots of pi-pi and cation-pi interactions. While the force fields used here can reproduce these interactions to some extent they don't include polarization, which is especially important in cation-pi interactions, and so the interaction energies can be inaccurate (see for example, DOI: 10.1021/acs.jctc.0c00637). Modified force fields such as CHARMM36-WYF and QM/MM calculations have been used elsewhere to improve such calculations. The results in this manuscript in general show quite a few cation-pi interactions (Supp Fig 5) as might be expected given the amino acid

content in these unusual DNAJB6 regions, with many observed in AR2 interactions and also the CTD-G/F1 interactions. It might be expected that the more likely risk in using unmodified force fields without polarization is not detecting some cation-pi interactions. Thus, for example, should this affect confidence in the result that the delta-ST leads to loss of the AR2-JD contacts, which is composed of many cation-pi interactions in WT? Limitations of the force field may also be relevant for the P96R results, which are an outlier in the contact maps (Supp Fig.12). This is perhaps not critical problem here but it may be worth a note in the methods or in the main text if it is pertinent to a particular conclusion.

How unique are the closed and open structural states? There is a description (lines 143-149) of partial transitions between conformations, potentially most interesting being a conversion from extended to a 'near-closed' state. How similar are the closed and near-closed states? This might be used as a proxy for assessing reversibility and confidence that these are actually on-pathway transition states would need to be adjusted depending on how similar the starting and ending states are.

Related to the above, how can we be sure that some of these conformations aren't simply unfolding intermediates – especially the extended state? One might expect a correlation with opening/extending and release of the helix V inhibition but that is not seen here (assuming it is not a limitation of the force fields and the helix V is artificially stuck to the JD). Of course, there is no substrate here but the fact that the JD/helix V interaction is present in all means there is no obvious basis for arguing that the extended structure is relevant. It is argued that it makes sense that the extended conformation is the active conformation because it exposes better the ST and the CTD. But it also uncouples the structured domains and results in rather large distances between the substrate binding regions and the helix V / JD, making it less obvious how substrate binding might be coupled to releasing inhibition. The open state, seems most compelling as an on-pathway conformation: (1) the substrate binding regions CTD and ST are exposed, (2) the structured regions are still interacting and so could provide an allosteric switch upon substrate binding, and (3) it is easier to imagine how the fixed orientation of the substrate binding CTD(/ST) relative to the JD is more efficient for transferring client proteins to the Hsp70 SBD when bound via the JD.

Lines 150-151: The Supp Fig. 6 text indicates that ST region did not intervene in 2 of 5 instances of closed-to-open transitions. Does this make it somewhat more unlikely than the main text implies that the G/F1-ST interaction is necessary for the transition?

A few questions regarding Fig 5a, the RMSF for WT and the LGMDD1 variants:

(1) It is unclear how to understand the y-axis scale. The RMSF seems to be a local measure, and one can see that for residues within helix V the RMSF is low and so presumably the helix is largely maintained, but these fluctuations are still on the order of 10Å, which seems very large. Is this reflecting helix V moving relative to the JD? Is molecular tumbling removed through a whole structure alignment, and if so does that cause problems if there are relative motions between the structured domains (e.g., JD and

CTD)? It would be helpful to see the RMSF for a larger region, certainly including the entire helix V, and perhaps in supplemental the RMSF across the entire protein.

(2) Is there an evaluation of whether the simulations have converged in the 2 μ s?

(3) It isn't explicitly stated that there was an initial energy minimization of the LGMDD1 structures after producing the variant structures. Is this done in VMD or otherwise, and if not, would any rearrangement of the structure from the WT conformation affect the RMSF values here? Presumably that could be confirmed by comparison after omitting some of the early simulation timepoints, and that might also help address question (2) above.

Overall, the contact persistence map results shown in Supplementary Fig.12 are much more direct and useful?

A minor point, in last paragraph of results section, it's stated that F91L shows a 'minor loss of function(LOF) in preventing polyQ aggregation...' but it seems there is no loss of function at all, at least in the data shown in Fig. 5b?

Reviewer #2 (Remarks to the Author):

This is a very good paper that reports the dynamics of the DNAJB6b chaperone in relation to its binding to Hsp70. This process has implications for muscular dystrophy and, consequently, the study can contribute to understanding the mechanism of this disease and to designing therapies. The authors have carried out microsecond-scale all-atom MD simulations to study the transition between the closed and open state of the DNAJ variant and its mutants (3 trajectories per each). The structure of the protein was modeled by using several state-of-the-art bioinformatics methods, including the recent AlphaFold and the most plausible one was selected based on the agreement with the results of companion XL-MS experiments. An extended intermediate, not reported previously, was found and transitions from this extended intermediate to both the closed and the extended state were observed. Molecular interactions that are responsible for structure formation and variations were discussed.

The paper is very well and comprehensibly written and the calculations and results analysis appear to be flawless. I recommend the publication of the manuscript as it is.

REPLY TO THE REVIEWER'S COMMENTS (reviewer comments in black, our response in blue)

Reviewer #1 (Remarks to the Author):

This is an intriguing paper using molecular dynamics simulations to try and understand potential intramolecular interactions of DNAJB6, which have been difficult to study by wet-lab approaches. The simulations raise interesting hypotheses about the ST region and CTD regulating access to the J-domain by Hsp70s, which may be linked to substrate binding. The hypothesis is very convincing in consideration of the biological context (inefficient to bind/sequester an Hsp70 unless there is substrate to deliver). It can be difficult however to be convinced by the specific details of the DNAJB6 structures and their interconversions in the absence of experimental validation (for example from H/D exchange mass spec or perhaps additional cross-linking mass spec). It was generally quite difficult to know which details of the models had high confidence and where the limits of interpretation might lie. At the same time, it is appreciated that it has been very difficult to get atomic resolution information on the native, full-length version of this protein, and the sort of structural framework proposed could be significant in moving the field forward with testable hypotheses.

We thank the reviewer for the positive comments on our study and its importance in pushing the field forward. We agree that experimental validation of the predicted interconversions within DNAJB6 would be highly complementary. However, as also stated by the reviewer, it is difficult to experimentally obtain atomic resolution data on the full-length version of this protein, especially because of its dynamic nature. Such an experimental exploration goes beyond the scope of the current manuscript, but we would like to highlight that some of our predictions (e.g. those on the LGMD mutants, related to the disruption of auto-inhibition) have been experimentally validated by NMR in the co-submitted paper of the Rosenzweig group, supporting the validity of our molecular dynamics simulations.

Confidence that the chosen starting structure ('closed') is relevant in terms of the JD/CTD arrangement, or even whether it is the most relevant from the list of structures in Supp Table 1 is not high considering the information provided. The closed structure is chosen from a set of structures from Robetta, RaptorX, TR-Rosetta, or AlphaFold and based on agreement with CLMS data. Firstly, none the structures is able to satisfy all the lysine crosslinks (in Robetta #4, K196-K29 is 40Å, well above the standard cut-off). In addition, K225 crosslinks K21 and K70 which are on opposite faces of the J-domain, and K196 and K225 (on one face of the CTD beta sheet) crosslink to K29 and K60, respectively, which are on opposite faces of the JD. These crosslinks suggest there is more than one docking pose and/or a high degree of flexibility. Second, the TR-Rosetta #1 structure matches the CLMS data as well as Robetta #4 (20.46 vs. 19.07). How different are TR-Rosetta #1 and Robetta #4 from each other, and are there other criteria that might justify using one over the other?

Related to the above, are there common differences between the entire set of structures that might give insights into potential conformational changes, which might complement the results from the MD simulations?

We appreciate the reviewers' comments related to the cross-link data, our choice for the starting structure and the possible insights to be gained from relating the proposed (homology) structures to our MD data. Indeed, DNAJB6's flexibility may be one of the key characteristics that allow it to perform its biological function, and we agree with the reviewer that this flexibility might be reflected in the crosslinking mass spectrometry (CLMS) data.

To select our starting configuration, we compared every structure generated by Robetta, TR-Rosetta, RaptorX and AlphaFold to the existing NMR structure of delta-S/T DNAJB6b [1] in terms of the secondary structure and radius of gyration (R_g) of the J+G/F domain. We found that all generated DNAJB6b structures matched well with the NMR structure and thus did not allow us to discriminate between the different models from the predictors. We therefore decided to aim for the lowest error in the cross-link data for full-length DNAJB6b, resulting in both Robetta #4 and TR-Rosetta #1 as possible candidates. Here, however, we observed a long stretch of beta-strand at the C-terminal of the S/T domain in all the TR-Rosetta models, which is in contradiction to the disordered nature of the S/T domain [1]. This led to our choice for Robetta #4. To make these considerations more explicit in the revised manuscript, we added the following text to the 1st paragraph of the section 'Initial configuration for the all-atom MD simulations': "Initially the models were compared to the existing NMR structure of Δ S/T DNAJB6b [24] in terms of secondary structure and the radius of gyration (R_g of the J+G/F domain) and it was found that all generated structures were closely aligned with the NMR structure."

Based on the question of the reviewer, we now also further compared the three predicted states of the MD simulations (closed, open, extended) to the structures generated by homology modelling. Interestingly, all three states predicted by the MD simulations were included in the list of homology structures: the highly extended conformation of the S/T domain was predicted by AlphaFold and the models generated by Robetta and TR-Rosetta predict a mix of closed (4) and open states (6). As also mentioned above, every model of TR-Rosetta predicts a beta-strand in the S/T domain, a feature not observed in any of the 6 simulations we performed, in line with [1]. It is rewarding to see that the dynamic and flexible nature of DNAJB6b in our simulations is well in line with the range of structures predicted by homology modelling, even though all simulations started from one specific geometry in the closed state. We have added these observations in the first paragraph of the Discussion section of the revised manuscript, stating: *“Interestingly, these three states were also found among the multiple states predicted by homology modelling: the highly extended conformation of the S/T domain was predicted by AlphaFold and the models generated by Robetta and TR-Rosetta predicted a mix of closed (4) and open states (6).”*

This is a system with lots of pi-pi and cation-pi interactions. While the force fields used here can reproduce these interactions to some extent they don't include polarization, which is especially important in cation-pi interactions, and so the interaction energies can be inaccurate (see for example, DOI: 10.1021/acs.jctc.0c00637). Modified force fields such as CHARMM36-WYF and QM/MM calculations have been used elsewhere to improve such calculations. The results in this manuscript in general show quite a few cation-pi interactions (Supp Fig 5) as might be expected given the amino acid content in these unusual DNAJB6 regions, with many observed in AR2 interactions and also the CTD-G/F1 interactions. It might be expected that the more likely risk in using unmodified force fields without polarization is not detecting some cation-pi interactions. Thus, for example, should this affect confidence in the result that the delta-ST leads to loss of the AR2-JD contacts, which is composed of many cation-pi interactions in WT? Limitations of the force field may also be relevant for the P96R results, which are an outlier in the contact maps (Supp Fig.12). This is perhaps not critical problem here but it may be worth a note in the methods or in the main text if it is pertinent to a particular conclusion.

We appreciate the comments regarding possible limitations of the force fields we used, specifically regarding the polarization in cation-pi interactions.

To make sure that our conclusions still hold for the loss of AR2 contacts in the delta-ST DNAJB6b system, we have now performed three additional simulations (three replicas of 2 μ s each) with the CHARMM36-WYF force field for the delta-ST DNAJB6b system. In the first replica, we identified the presence of AR2 contacts, even though they were not as pronounced as in the full-length DNAJB6b simulations, while in the second and third replicas we did not observe them. Similarly, for the first replica we noted some HPD contacts with the CTD, but in the second and third replicas these contacts were absent. Based on this we conclude that there can be some AR2 and HPD-CTD contacts in delta-ST DNAJB6b but much less pronounced than in full-length DNAJB6b. We have made the following changes to the main and supplementary text to incorporate these additional simulations:

1. We updated the contact maps in Figs. 4d, h and we modified the main text with reference to these figures in the second paragraph of the section ‘DNAJB6b: autoinhibition via anchor region 1’, stating: *“Although we observed some AR2 interactions, these were not as prominent as in the full-length DNAJB6b.”*
2. Supplementary Figs. 14a, b and c are updated including the caption.
3. Simulation details are added in the last paragraph of the subsection ‘Molecular dynamics simulations’ in the section ‘Methodology’.

Concerning the implications of the LGMDD1 mutation P96R: In our study, we found that the mutated R96 forms a cation-pi interaction with F100, thus competing with F100 to interact with the J-domain, ultimately leading to the loss of autoinhibition. Considering that additional polarization will lead to stronger bonding, a cation-pi model such as CHARMM36-WYF will most likely predict a similar outcome, with (strong) cation-pi interactions being an indirect cause for the loss of autoinhibition.

How unique are the closed and open structural states? There is a description (lines 143-149) of partial transitions between conformations, potentially most interesting being a conversion from extended to a 'near-closed' state. How similar are the closed and near-closed states? This might be used as a proxy for assessing reversibility and confidence that these are actually on-pathway transition states would need to be adjusted depending on how similar the starting and ending states are.

The closed and near-closed states are very similar and the same holds for the open and near-open states as shown in Supplementary Fig. 10. In the near-closed state, the CTD does not yet interact with the G/F₁ domain but interacts with the N-terminal of the G/F₂ domain instead, while there are fewer interactions between the S/T domain and the G/F₂ domain compared to the closed state. Considering these small differences, it can be expected that, with time, the near-closed state will transition to the closed state, making the near-closed state to represent an on-pathway transition from the intermediate conformation to the closed state. Similarly, the near-open state (Supplementary Fig. 10d) only differs from the open state by a compression of the S/T domain. Considering this small difference, it is reasonable to expect that, with time, the S/T domain will collapse, thus making the near-open state an on-pathway transition to the open state. The following changes were made in the main and supplementary text to incorporate this:

1. In the main-text, we now clearly mention that the near-closed and near-open states are interpreted as on-pathway transitions to the closed and open states, respectively. This modified text is in the 2nd paragraph of the section 'Conformational dynamics of the cycling of DNAJB6b between the closed, open and extended state', stating: *"....after which it transitioned to a near-closed state, that can be interpreted as an on-pathway transition to the closed state (see Supplementary Fig. 10c)." and "... further back to the near-open state, suggesting an on-pathway transition to the open state (Supplementary Fig. 10d) ..."*.
2. In the supplementary text, we have changed the supplementary Fig. 10 and its caption, now explicitly referring to the states after the intermediate conformation as near-closed and near-open states.

Related to the above, how can we be sure that some of these conformations aren't simply unfolding intermediates – especially the extended state?

Based on the contact maps obtained from six independent simulations of full-length DNAJB6b, we observed that JB6b almost exclusively samples three states: closed, open and extended (supplemented by short-lived intermediate states). This observation, together with the fact that these three states were also found in the homology modelling, made us conclude that DNAJB6b dynamically cycles between these three states and that these states are not unfolding states. We have added these observations in the last paragraph of the section 'Conformation dynamics of the cycling of DNAJB6b between the closed, open and extended state', stating: *"Hence, based on six independent simulations, combined with the observation that these states were also found in the homology models of Robetta and TR-Rosetta, we conclude that monomeric DNAJB6b converges to a dynamic ensemble of three interconverting states, cyclically changing between closed, open and extended."*

One might expect a correlation with opening/extending and release of the helix V inhibition but that is not seen here (assuming it is not a limitation of the force fields and the helix V is artificially stuck to the JD). Of course, there is no substrate here but the fact that the JD/helix V interaction is present in all means there is no obvious basis for arguing that the extended structure is relevant.

It is argued that it makes sense that the extended conformation is the active conformation because it exposes better the ST and the CTD. But it also uncouples the structured domains and results in rather large distances between the substrate binding regions and the helix V / JD, making it less obvious how substrate binding might be coupled to releasing inhibition. The open state, seems most compelling as an on-pathway conformation: (1) the substrate binding regions CTD and ST are exposed, (2) the structured regions are still interacting and so could provide an allosteric switch upon substrate binding, and (3) it is easier to imagine how the fixed orientation of the substrate binding CTD(/ST) relative to the JD is more efficient for transferring client proteins to the Hsp70 SBD when bound via the JD.

The work of Kuiper et al. [2], highlighted the critical role of the phenylalanine residues in the S/T domain in preventing the aggregation of FG-Nup fragments. In our simulations, these phenylalanine (F) residues predominantly interact with the G/F₂ domain in both the open and closed states (Fig. 2d, 2e), and are only exposed in the extended state (Fig. 2f). Additionally, serine (S) and threonine (T) residues, which are crucial for preventing polyglutamine aggregation [5,7], are partially buried in the open and closed state and only become highly exposed in the extended state. Considering that these key residues (phenylalanine (F), serine (S), and threonine (T)) are only (highly) exposed in the extended state, we propose that the extended state may represent the functional state of the protein. In the revised Discussion we now explain this in more detail in the 4th paragraph, stating: *"The work of Kuiper et al. [16], highlighted the critical role of the phenylalanine residues*

in the S/T domain in the ability of DNAJB6 to suppress the aggregation of FG-Nup fragments. In our simulations, these phenylalanine (F) residues predominantly interact with the G/F₂ domain in both the open and closed states (Fig. 2d, 2e), and are only exposed in the extended state (Fig. 2f). Since the serine (S) and threonine (T) residues of the S/T region are crucial for DNAJB6b to chaperone polyQ [9] and A β [11] and the phenylalanine (F) residues of the S/T domain are crucial for preventing the aggregation of FG-Nup fragments [16], the enhanced exposure of the S, T and F residues renders the extended state to be highly efficient compared to both open and closed states.”

The fact that auto-inhibition is maintained in all three observed states, tempted us to speculate that auto-inhibition is released by means of an allosteric mechanism, as also proposed for DNAJB8 [6]. We speculate that the substrate will be captured by the DNAJB6b in the extended state, and an allosteric effect will be triggered by the substrate binding which in turn opens helix 2/3 for interacting with Hsp70. However, we acknowledge that the molecular details on how this mechanism might lead to a compaction of the S/T-substrate complex to overcome the rather large distances between the substrate binding region and the helix V / JD are unknown at this stage and that future experiments or simulations must be carried out to clarify this. We added these considerations to paragraph 2 of the revised Discussion, stating: *“However, the molecular details on how this mechanism can lead to compaction of the S/T-substrate complex in order to transfer the substrate to HSP70 at the J-domain is unknown at this stage.”*

Lines 150-151: The Supp Fig. 6 text indicates that ST region did not intervene in 2 of 5 instances of closed-to-open transitions. Does this make it somewhat more unlikely than the main text implies that the G/F₁-ST interaction is necessary for the transition?

We appreciate for bringing this point to our attention. Upon reflection, we agree that our original wording may have suggested a more definitive role for the G/F₁-S/T interaction in the closed-to-open transition than warranted by the evidence. To accommodate this, we have rephrased the main text in the first paragraph of the section ‘Conformational dynamics of the cycling of DNAJB6b between the closed, open and extended state’, stating the following: *“In three out of five simulations, the transition from closed to open relied on the competition between the S/T domain and the CTD for interactions with the G/F₁ domain (Supplementary Fig. 6, 7) and in the other two simulations, we observed that the fluctuations in the CTD conformations alone were enough for the CTD to be released from the G/F₁ domain thus transitioning to the open state.”*

A few questions regarding Fig 5a, the RMSF for WT and the LGMDD1 variants:

(1) It is unclear how to understand the y-axis scale. The RMSF seems to be a local measure, and one can see that for residues within helix V the RMSF is low and so presumably the helix is largely maintained, but these fluctuations are still on the order of 10Å, which seems very large. Is this reflecting helix V moving relative to the JD? Is molecular tumbling removed through a whole structure alignment, and if so does that cause problems if there are relative motions between the structured domains (e.g., JD and CTD)? It would be helpful to see the RMSF for a larger region, certainly including the entire helix V, and perhaps in supplemental the RMSF across the entire protein.

We thank the reviewer for this valuable observation. We first removed all the translational and rotational motion of the molecule using gmx trjconv. Then, we carried out an alignment of the entire molecule, and used the average structure of the open state of WT-DNAJB6b as a reference for computation of the RMSF values. This observation thus shows the dynamic nature of this system, especially the terminal residues of the CTD fluctuations relative to the JD. In response to the reviewer’s suggestion, we have extended our RMSF analysis to encompass the entire protein. The results of this expanded analysis are now included in the revised manuscript and it can be seen that the RMSF values of the CTD are relatively large (Supplementary Fig. 13). Additionally, when we performed alignment focusing on the less fluctuating J-domain and helix V, the RMSF trend for the G/F₁ linker remained consistent and the RMSF values of the J-domain and helix V are notably low. This suggests that the helix V is stable and it did not lose its secondary structure throughout our simulations relative to the J-domain. Importantly, the core conclusions made in our original MS remain unaffected. In the modified manuscript, we have updated Fig. 5a and added the RMSF of the entire protein and RMSF of residues 1- 104 with alignment on the J-domain and helix V in the supplementary Fig. 13. We have also updated the 3rd paragraph of the Methodology section named ‘Analysis of the MD simulations’ to explain how we computed the RMSF.

(2) Is there an evaluation of whether the simulations have converged in the 2 μ s?

Given the dynamic nature of DNAJB6, traditional metrics such as RMSD and Rg will not provide an accurate picture of convergence. This is primarily because the protein transitions between different states, which can significantly alter these values. In our study, we observed sampling into three predominant states (open, closed, and extended) during the course of the six independent simulations of 2 μ s. So instead of convergence to a single static state, we observed convergence to a dynamic ensemble of three interconverting states. To better reflect this in the revised manuscript, we have modified the text in the last three lines of the last paragraph in the section 'Conformational dynamics of the cycling of DNAJB6b between the closed, open and extended state', stating: "Hence, based on six independent simulations, combined with the observation that these states were also found in the homology models of Robetta and TR Rosetta we conclude that monomeric DNAJB6b converges to a dynamic ensemble of three interconverting states, cyclically changing between closed, open and extended."

(3) It isn't explicitly stated that there was an initial energy minimization of the LGMDD1 structures after producing the variant structures. Is this done in VMD or otherwise, and if not, would any rearrangement of the structure from the WT conformation affect the RMSF values here? Presumably that could be confirmed by comparison after omitting some of the early simulation timepoints, and that might also help address question (2) above.

We indeed performed energy minimization on the LGMDD1 structures after introducing the mutations. This was equal to the simulation protocol of the wildtype DNAJB6b simulations, as mentioned in the last line of the third paragraph of the section 'Molecular dynamics simulations' in the Methodology section. To make this clearer we now explicitly mention this at the end of this section in the revised manuscript.

Overall, the contact persistence map results shown in Supplementary Fig.12 are much more direct and useful?

We agree that these maps can provide insightful information and we indeed considered this, but we decided to include the RMSF plots in the main text instead because of its more compact representation and straightforward interpretation by a broader audience.

A minor point, in last paragraph of results section, it's stated that F91L shows a 'minor loss of function (LOF) in preventing polyQ aggregation...' but it seems there is no loss of function at all, at least in the data shown in Fig. 5b?

The reviewer is correct. While for some LGMD-related mutants a mild LOF was found for suppressing polyQ aggregation [3,4], here the functionality of the F91L mutant was statistically not significantly different from that of the wildtype protein. Why this is slightly different for different mutants is unclear and might depend on the different experimental conditions due to the fact that for this effect in cells, the holdase activity of DNAJB6 is rate-limiting. This is not only supported by the data showing that the H/Q variant also has minor impact on the suppression of polyQ aggregation in cells (see [5] and this manuscript), but supported by the in vitro data from the paper submitted in parallel by the Rosenzweig group. In fact, the dependency of Hsp70 interactions (for both the H/Q variant and the F91L mutant) only becomes evident when we slightly impair this holdase activity of DNAJB6, consistent with the findings that interaction with Hsp70 is required for processing the DNAJB6-bound polyQ substrate [5] and supporting the idea that the problem with the LGMD-related DNAJB6 mutants is not in the holdase activity but in proper connections to the Hsp70 machine. We have adapted our text accordingly and changes are reflected in the last paragraph of the subsection 'Spontaneous loss of autoinhibition in DNAJB6b is a feature of LGMDD1-associated mutations' in the Results section, now stating the following: "To extrapolate this to functionality, we turned to cellular experiments and addressed how LGMDD1 mutants of DNAJB6b might affect its anti-aggregation properties towards polyglutamine proteins. It was shown before [6, 18] that some LGMDD1 mutants show a minor loss of function (LOF) in preventing polyQ aggregation compared to wildtype DNAJB6b. However, we did not detect a functional defect in the DNAJB6b-F91L mutant (Fig. 5b, 5c), which might either be due to different severity of the F91L mutant or to the experimental setup of the present study. However, it supports the notion that the G/F 1 region (with LGMD mutants) is not critical for the 'holdase' function required primarily in preventing polyQ aggregation in cells [7, 9]. It is also in line with recent in vitro data showing that LGMD mutants are not affecting the substrate binding and their 'holdase' capacity to suppress aggregation of polyQ and TDP43 [32]. In fact, the same was observed in cells for the DNAJB6b-H/Q mutant that has a fully intact substrate binding domain, but that cannot interact with Hsp70 [7, 9]. The impact of this H/Q mutation, i.e.,

improper interaction with Hsp70 is only revealed when combined with a second mutant that alone slightly impairs this substrate binding and anti-polyQ aggregation capacity due to a replacement of five of the serine (S) and threonines (T) of the substrate binding domain into alanines (6 S/T > A). This double mutant now shows a complete loss of function [9]. When combining the F91L mutant with such a second substrate binding impairing mutation, we also find more than additive effects and the double mutant (H/Q + F91L) is strongly impaired in preventing polyQ aggregation (Fig. 5b). In contrast, introducing a H/Q mutation in the background of F91L mutant show no additive effects (Fig. 5b) as they both affect Hsp70 interactions.

Reviewer #2 (Remarks to the Author):

This is a very good paper that reports the dynamics of the DNAJB6b chaperone in relation to its binding to Hsp70. This process has implications for muscular dystrophy and, consequently, the study can contribute to understanding the mechanism of this disease and to designing therapies. The authors have carried out microsecond-scale all-atom MD simulations to study the transition between the closed and open state of the DNAJ variant and its mutants (3 trajectories per each). The structure of the protein was modeled by using several state-of-the-art bioinformatics methods, including the recent AlphaFold and the most plausible one was selected based on the agreement with the results of companion XL-MS experiments. An extended intermediate, not reported previously, was found and transitions from this extended intermediate to both the closed and the extended state were observed. Molecular interactions that are responsible for structure formation and variations were discussed.

The paper is very well and comprehensibly written and the calculations and results analysis appear to be flawless. I recommend the publication of the manuscript as it is.

We appreciate the very positive evaluation by this reviewer.

References used in this rebuttal

- [1]. Söderberg, Christopher AG, Cecilia Månsson, Katja Bernfur, Gudrun Rutsdottir, Johan Härmark, Sreekanth Rajan, Salam Al-Karadaghi et al. "Structural modelling of the DNAJB6 oligomeric chaperone shows a peptide-binding cleft lined with conserved S/T-residues at the dimer interface." *Scientific reports* 8, no. 1 (2018): 5199.
- [2]. Kuiper, EF Elsiena, Paola Gallardo, Tessa Bergsma, Muriel Mari, Maiara Kolbe Muszkopf, Jeroen Kuipers, Ben NG Giepmans et al. "The chaperone DNAJB6 surveils FG-nucleoporins and is required for interphase nuclear pore complex biogenesis." *Nature Cell Biology* 24, no. 11 (2022): 1584-1594.
- [3]. Sarparanta, Jaakko, Per Harald Jonson, Christelle Golzio, Satu Sandell, Helena Luque, Mark Screen, Kristin McDonald et al. "Mutations affecting the cytoplasmic functions of the co-chaperone DNAJB6 cause limb-girdle muscular dystrophy." *Nature genetics* 44, no. 4 (2012): 450-455.
- [4]. Thiruvalluvan, Arun, Eduardo P. de Mattos, Jeanette F. Brunsting, Rob Bakels, Despina Serlidaki, Lara Barazzuol, Paola Conforti et al. "DNAJB6, a key factor in neuronal sensitivity to amyloidogenesis." *Molecular cell* 78, no. 2 (2020): 346-358.
- [5]. Hageman, Jurre, Maria A. Rujano, Maria AWH Van Waarde, Vaishali Kakkar, Ron P. Dirks, Natalia Govorukhina, Henderika MJ Oosterveld-Hut, Nicolette H. Lubsen, and Harm H. Kampinga. "A DNAJB chaperone subfamily with HDAC-dependent activities suppresses toxic protein aggregation." *Molecular cell* 37, no. 3 (2010): 355-369.
- [6]. Ryder, Bryan D., David R. Boyer, Elizaveta Ustyantseva, Ayde Mendoza-Oliva, Mikolaj Kuska, Pawel M. Wydorski, Michael Sawaya et al. "DNAJB8 oligomerization is mediated by an aromatic-rich motif that is dispensable for substrate activity." *bioRxiv* (2023): 2023-03.
- [7]. Kakkar, Vaishali, Cecilia Månsson, Eduardo P. de Mattos, Steven Bergink, Marianne van Der Zwaag, Maria AWH van Waarde, Niels J. Kloosterhuis et al. "The S/T-rich motif in the DNAJB6 chaperone delays polyglutamine aggregation and the onset of disease in a mouse model." *Molecular cell* 62, no. 2 (2016): 272-283.

REVIEWER COMMENTS

Reviewer #1 (Remarks to the Author):

The authors' responses are helpful in evaluating confidence in the structures and interactions observed in the simulations. I think the additions and clarifications to the text have contributed to a much improved manuscript. While I do think this is an important contribution to the field with regards to providing a hypothetical framework for thinking about the mechanisms underlying DNAJB6 regulation and activity I continue to have concerns with some conclusions that are too strongly worded.

I don't find that the simulation data sufficiently supports the conclusion that the LGMDD1 substitutions release autoinhibition. It seems that the helix V spends little or no time dissociated in the simulations even for the two mutants within the helix V that are at the JD interface (F100I and F100V). Fig 5 shows that Helix V in P96R was seen to dissociate but the significance of this is unclear; was this observed just this once? Was it seen in any other simulations? Instead, changes in RMSFs in the linker region are taken to indicate a loosening of the autoinhibition but even those changes in RMSFs are not convincing. Firstly, the Supp Fig 13 indicates the RMSFs for helix V when structures are aligned with the JD are actually lower than WT for F91L and F93L. One might expect larger Helix V RMSFs for the F100 mutations considering they are within Helix V and pack against the JD but the helix V RMSFs don't seem significantly elevated above WT. In addition, the elevated RMSFs in the linker for the mutations could be a trivial result: as might be expected, mutations within the linker (F91L, F93L, P96R; all of which are expected to increase dynamics due to reduced bulkiness or removal of a proline) have the strongest effects on linker dynamics whereas those that are not in the linker (F100I/V mutants) have less effect on the linker dynamics. But critically, this does not correlate with the rank order of helix V dissociation seen in experimental NMR work. Moreover, it is not clear why these RMSF calculations are focused on the open state; the contact maps in Supp Fig. 12 suggests some differences between the open and closed state, and there is no a priori information indicating what is the relevant conformation. In summary, it is not clear that the simulations are sensitive enough to support the mechanistic conclusions stated multiple times about the LGMDD1 mutations (for example, lines 294-295: "Our MD data suggest that these LGMD mutants can drive the spontaneous release of autoinhibition.", and lines 347-349 "The importance of this regulation is highlighted by the finding that Hsp70 access is restricted in all non-substrate loaded DNAJB6b monomers and that mutants that cause LGMD relieve this restriction."). The MD simulations on their own are far from clear in this regard and I'm not sure the same conclusions would be made if the results of the experimental work done elsewhere were not known.

I apologize that I did not pick up on the following the first time through but below are some additional, relatively minor points:

Lines 256-258 (tracked changes version): Considering that the starting models for the simulations had helix V docked against the JD, the statement that MD simulations confirm experimental NMR data in regards to the auto-inhibited state seen experimentally by Karamanos et al. (and now also Abayev-Avraham et al.) seems too strong. Perhaps this could be concluded from the MD simulations if it was observed that the undocked structure became docked during the course of the simulations. It is also perhaps not quite fair to refer to the delta-S/T mutant used by Karamanos et al. as a 'fragment of DNAJB6b' (line 258 in tracked changes version).

Lines 285-288 (tracked changes version): Comparison of the apparent K_d for homo-oligomerization of DNAJB6b and the concentration needed to observe DNAJB6b in vitro holdase activity suggests that monomeric DNAJB6b is active and so able to bind substrate (for example Linse 2022 and Mansson 2018), and it seems like an unneeded complication to invoke the requirement for oligomerization to then release auto-inhibition?

Lines 329-334 (tracked changes version): It should be pointed out that Ryder et al. 2023 tested the role of those DNAJB6b residues equivalent to the ones in DNAJB8 that are involved in oligomerization and did not see monomerization of DNAJB6b; they concluded that a different mechanism was occurring in DNAJB6b.

Typo in line 280 (tracked changes version) "in which ~relieve~ of autoinhibition".

REPLY TO THE REVIEWER'S COMMENTS (reviewer comments in black, our response in blue)

Reviewer #1 (Remarks to the Author):

The authors' responses are helpful in evaluating confidence in the structures and interactions observed in the simulations. I think the additions and clarifications to the text have contributed to a much improved manuscript. While I do think this is an important contribution to the field with regards to providing a hypothetical framework for thinking about the mechanisms underlying DNAJB6 regulation and activity I continue to have concerns with some conclusions that are too strongly worded.

We thank the reviewer for the positive words. Below we respond to the specific remarks of the reviewer and revisited critically the wording used to state the conclusions.

I don't find that the simulation data sufficiently supports the conclusion that the LGMDD1 substitutions release autoinhibition. It seems that the helix V spends little or no time dissociated in the simulations even for the two mutants within the helix V that are at the JD interface (F100I and F100V).

The reviewer is correct in stating that the F100I and F100V mutations did not result in complete destabilization of the autoinhibition and we fully agree. This is due to residues F103 and F104 that partially compensate for F100 in the interaction with the J-domain. This is pointed out in the last line of the second paragraph of the section "Spontaneous loss of autoinhibition in DNAJB6b is a feature of LGMDD1-associated mutations."

Fig 5 shows that Helix V in P96R was seen to dissociate but the significance of this is unclear; was this observed just this once? Was it seen in any other simulations?

We have performed six simulations per mutation and complete destabilization of autoinhibition is observed for the P96R mutation in one simulation: this event happened at ~1700 ns and the destabilization remained for the rest of the simulation (+300 ns). We have now incorporated this information in the section "Spontaneous loss of autoinhibition in DNAJB6b is a feature of LGMDD1-associated mutations":

Line 209-211: "In one of the P96R mutation simulations, we did observe a complete destabilization of the autoinhibitory state at around ~1700 ns which remained destabilized for the duration of the simulation (an additional 300 ns)."

Line 218 – 219: "The destabilization of autoinhibition for the P96R mutant, together with the induced fluctuations of the residues in the G/F₁ linker for all the LGMD mutants, suggest that LGMD-causing mutations in the G/F₁ domain may have an impact on the stability of the autoinhibitory state...."

Instead, changes in RMSFs in the linker region are taken to indicate a loosening of the autoinhibition but even those changes in RMSFs are not convincing. Firstly, the Supp Fig 13 indicates the RMSFs for helix V when structures are aligned with the JD are actually lower than WT for F91L and F93L.

In Supp Fig 13b, prior to the calculation of the RMSF, alignment was performed on both the J domain + helix V (residues 1-74, 96-104). We found that the higher RMSFs for helix V of F91L and F93L are a consequence of this specific choice of the alignment. Since the protein is very dynamic, it is difficult to obtain a proper alignment that is representative for the full simulation trajectory. Consequently, the higher RMSFs are due to this improper alignment. In order not to bias observations based on alignment, we have removed Supp Fig. 13b from the supplementary info.

One might expect larger Helix V RMSFs for the F100 mutations considering they are within Helix V and pack against the JD but the helix V RMSFs don't seem significantly elevated above WT. In addition, the elevated RMSFs in the linker for the mutations could be a trivial result: as might be expected, mutations within the linker (F91L, F93L, P96R; all of which are expected to increase dynamics due to reduced bulkiness or removal of a proline) have the strongest effects on linker dynamics whereas those that are not in the linker (F100I/V mutants) have less effect on the linker dynamics.

Yes, this might be the case. As explained in our second response, we think that the lower effect on the linker dynamics is due to the partial compensation from residues F103 and F104 of helix V.

But critically, this does not correlate with the rank order of helix V dissociation seen in experimental NMR work. Moreover, it is not clear why these RMSF calculations are focused on the open state; the contact maps in Supp Fig. 12 suggests some differences between the open and closed state, and there is no a priori information indicating what is the relevant conformation.

In the closed state, the CTD interacts with the G/F₁ domain and we do not expect destabilization of autoinhibition to occur in the more stable closed state. This is mentioned in the 3rd paragraph of the section “Spontaneous loss of autoinhibition in DNAJB6b is a feature of LGMDD1-associated mutations.”

In summary, it is not clear that the simulations are sensitive enough to support the mechanistic conclusions stated multiple times about the LGMDD1 mutations (for example, lines 294-295: “Our MD data suggest that these LGMD mutants can drive the spontaneous release of autoinhibition.”, and lines 347-349 “The importance of this regulation is highlighted by the finding that Hsp70 access is restricted in all non-substrate loaded DNAJB6b monomers and that mutants that cause LGMD relieve this restriction.”). The MD simulations on their own are far from clear in this regard and I’m not sure the same conclusions would be made if the results of the experimental work done elsewhere were not known.

Although in the P96R LGMD mutant we observed one case of complete mechanistic destabilization of autoinhibition and we observed a trend in the other LGMD mutants tested that the fluctuations of the G/F₁ linker were increased, we agree with the reviewer that the MD simulations do not unequivocally show a destabilisation of all point mutations within the time frame of the simulations. We therefore refrained from drawing too strong conclusions on the MD results alone, but refer to both MD and NMR data (Rosenzweig group) when concluding on release of auto-inhibition. To do so, following the reviewer’s suggestion, we have deleted lines 294-295 (old numbering) and added instead (lines 296-297): “Within the time frame of the MD simulations, we observed one complete destabilization event for the P96R mutant” and lines 303-304: “Combined, these findings suggest a spontaneous release of auto-inhibition upon LGMD mutations.” Finally, we changed lines 347-349 (old numbering) into lines 350-352: “The importance of this regulation is highlighted by the finding that Hsp70 access is restricted in all non-substrate loaded DNAJB6b monomers (our MD data) and that mutants that cause LGMD relieve this restriction (Abayev-Avraham et al. [32], our MD data).”

I apologize that I did not pick up on the following the first time through but below are some additional, relatively minor points:

Lines 256-258 (tracked changes version): Considering that the starting models for the simulations had helix V docked against the JD, the statement that MD simulations confirm experimental NMR data in regards to the auto-inhibited state seen experimentally by Karamanos et al. (and now also Abayev-Avraham et al.) seems too strong. Perhaps this could be concluded from the MD simulations if it was observed that the undocked structure became docked during the course of the simulations. It is also perhaps not quite fair to refer to the delta-S/T mutant used by Karamanos et al. as a ‘fragment of DNAJB6b’ (line 258 in tracked changes version).

We agree with the comments of the reviewer. We have made changes to the discussion section reflecting this, see our response to the previous remark.

We have rephrased “fragment of DNAJB6b” to “a DNAJB6b construct lacking the S/T domain” (Line 259).

The modified text now says:

Line 256-259 – “Similar to the DNAJB1-like proteins [27], the G/F region contains a helix V that block access of Hsp70 to the J-domain (autoinhibition), shown by the NMR data of Karamanos et al. [24] using a DNAJB6b construct lacking the S/T domain.”

Lines 285-288 (tracked changes version): Comparison of the apparent K_d for homo-oligomerization of DNAJB6b and the concentration needed to observe DNAJB6b in vitro holdase activity suggests that monomeric DNAJB6b

is active and so able to bind substrate (for example Linse 2022 and Mansson 2018), and it seems like an unneeded complication to invoke the requirement for oligomerization to then release auto-inhibition?

We think that the holdase activity of DNAJB6b doesn't require the autoinhibition to be released and, moreover, it is not yet shown that the substrate binding releases the autoinhibition for DNAJB6b. Therefore, we thought that the release of auto-inhibition by the allosteric effect of the self-oligomerization of DNAJB6b is also a possibility. We understand, however, that it is a unneeded complication and therefore removed lines 285-288 (old numbering) from the discussion.

Lines 329-334 (tracked changes version): It should be pointed out that Ryder et al. 2023 tested the role of those DNAJB6b residues equivalent to the ones in DNAJB8 that are involved in oligomerization and did not see monomerization of DNAJB6b; they concluded that a different mechanism was occurring in DNAJB6b.

We agree with the reviewer and made appropriate changes in the discussion section of the main text: the modified text now says:

Line 332-340 – “Another still open question relates to the role of self-oligomerization of the DNAJB6-subgroup. A recent study on DNAJB8 revealed that this is related to a specific peptide (¹⁴⁷AFSSFN¹⁵²) within the S/T rich region [53]. It was found that for DNAJB8, mutations in this peptide did abolish self-oligomerization and lead to DNAJB8 monomerization; this monomeric DNAJB8, however, did retain wildtype activity in terms of substrate binding and chaperone function [53]. We speculate that for DNAJB6b, substrate binding and self-oligomerization could be driven by the same S/T domain of DNAJB6b, and that monomers are the functional chaperone moieties, and self-oligomerization may only happen at high concentrations in the absence of clients.”

Typo in line 280 (tracked changes version) “in which ~relieve~ of autoinhibition”.

Typo was now corrected in the line 281 - “...relief...”